



# Cloud drop number concentrations over the western North Atlantic Ocean: seasonal cycle, aerosol interrelationships, and other influential factors

Hossein Dadashazar[1], David Painemal[2,3], Majid Alipanah[4], Michael Brunke[5], Seethala Chellappan[6],
Andrea F. Corral[1], Ewan Crosbie[2,3], Simon Kirschler[7], Hongyu Liu[8], Richard H. Moore[2], Claire Robinson[2,3],
Amy Jo Scarino[2,3], Michael Shook[2], Kenneth Sinclair[9,10], K. Lee Thornhill[2], Christiane Voigt[7], Hailong Wang[11],
Edward Winstead[2,3], Xubin Zeng[5], Luke Ziemba[2], Paquita Zuidema[6], and Armin Sorooshian[1,5]

[1]Department of Chemical and Environmental Engineering, University of Arizona, Tucson, AZ, USA
[2]NASA Langley Research Center, Hampton, VA, USA
[3]Science Systems and Applications, Inc., Hampton, VA, USA
[4]Department of Systems and Industrial Engineering, University of Arizona, Tucson, AZ, USA
[5]Department of Hydrology and Atmospheric Sciences, University of Arizona, Tucson, AZ, USA
[6]Rosenstiel School of Marine and Atmospheric Science, University of Miami, Miami, FL, USA
[7]Institute of Atmospheric Physics, German Aerospace Center, Oberpfaffenhofen, Germany
[8]National Institute of Aerospace, Hampton, VA, USA
[9]NASA Goddard Institute for Space Studies, New York, NY, USA
[10]Universities Space Research Association, Columbia, MD, USA
[11]Atmospheric Sciences and Global Change Division, Pacific Northwest National Laboratory, Richland, WA, USA

**Correspondence:** Hossein Dadashazar (hosseind@arizona.edu)

**Abstract.** Cloud drop number concentrations ($N_d$) over the western North Atlantic Ocean (WNAO) are generally highest during the winter (DJF) and lowest in summer (JJA), in contrast to aerosol proxy variables (aerosol optical depth, aerosol index, surface aerosol mass concentrations, surface cloud condensation nuclei (CCN) concentrations) that generally peak in spring (MAM) and JJA with minima in DJF. Using aircraft, satellite remote sensing, ground-based in situ measurement data, and reanalysis data, we characterize factors explaining the divergent seasonal cycles and furthermore probe into factors influencing $N_d$ on seasonal timescales. The results can be summarized well by features most pronounced in DJF, including features associated with cold-air outbreak (CAO) conditions such as enhanced values of CAO index, planetary boundary layer height (PBLH), low-level liquid cloud fraction, and cloud-top height, in addition to winds aligned with continental outflow. Data sorted into high- and low-$N_d$ days in each season, especially in DJF, revealed that all of these conditions were enhanced on the high-$N_d$

days, including reduced sea level pressure and stronger wind speeds. Although aerosols may be more abundant in MAM and JJA, the conditions needed to activate those particles into cloud droplets are weaker than in colder months, which is demonstrated by calculations of the strongest (weakest) aerosol indirect effects in DJF (JJA) based on comparing $N_d$ to perturbations in four different aerosol proxy variables (total and sulfate aerosol optical depth, aerosol index, surface mass concentration of sulfate). We used three machine learning models and up to 14 input variables to infer about most influential factors related to $N_d$ for DJF and JJA, with the best performance obtained with gradient-boosted regression tree (GBRT) analysis. The model results indicated that cloud fraction was the most important input variable, followed by some combination (depending on season) of CAO index and surface mass concentrations of sulfate and organic carbon. Future work is recommended to further understand aspects uncovered here such as impacts of free tropospheric aerosol entrainment on clouds, degree of boundary layer coupling,

wet scavenging, and giant CCN effects on aerosol–$N_d$ relationships, updraft velocity, and vertical structure of cloud properties such as adiabaticity that impact the satellite estimation of $N_d$.

## 1   Introduction

Aerosol indirect effects remain the dominant source of uncertainty in estimates of total anthropogenic radiative forcing (Boucher et al., 2013; Myhre et al., 2013). Central to these effects is knowledge about cloud drop number concentration ($N_d$), as it is the connection between the subset of particles that activate into drops (cloud condensation nuclei, CCN) and cloud properties. It is widely accepted that warm clouds influenced by higher number concentrations of aerosol particles have elevated $N_d$ and smaller drops (all else held fixed), resulting in enhanced cloud albedo at fixed liquid water path (Twomey, 1977) and potentially suppressed precipitation (Albrecht, 1989) and increased vulnerability to overlying air resulting from enhanced cloud-top entrainment (Ackerman et al., 2004).

Reducing uncertainty in how aerosols and clouds interact within a given meteorological context requires accurate estimates of $N_d$ and aerosol concentrations and properties. Since intensive field studies struggle to obtain broad spatial and temporal coverage of such data, satellite remote sensing and reanalysis datasets are relied on for studies examining intra- and interannual features over large spatial areas. Limitations of satellite retrievals are important to recognize. $N_d$ is not directly retrieved but derived using other parameters (e.g., cloud optical depth, cloud drop effective radius, cloud-top temperature) and with assumptions about cloud adiabatic growth and $N_d$ being vertically constant (Grosvenor et al., 2018). Aerosol number concentrations are usually represented by a columnar parameter such as aerosol optical depth (AOD) and thus not directly below clouds, which is the aerosol layer most likely to interact with the clouds. Furthermore, aerosol data are difficult to retrieve in cloudy columns. Reanalysis datasets circumvent issues for the aerosol parameters as they provide vertically resolved data (e.g., surface layer and thus below clouds) and are available for cloudy columns.

Of special interest in this work is the western North Atlantic Ocean (WNAO) where decades of extensive research have been conducted for topics largely unrelated to aerosol–cloud interactions (Sorooshian et al., 2020), thereby providing an opportunity for closing knowledge gaps for this area in a region with a wide range of aerosol and meteorological conditions (Corral et al., 2021; Painemal et al., 2021). Past work showed different seasonal cycles of AOD and $N_d$ in this region (Grosvenor et al., 2018; Sorooshian et al., 2019), which partly motivates this study to unravel why $N_d$ behaves differently on seasonal timescales. A previous study investigating seasonal cycles of $N_d$ in the North Atlantic region found that cloud microphysical properties were primarily dependent on CCN concentrations while cloud macrophysical properties were more dependent on meteorological conditions (e.g., Sinclair et al., 2020). However, due to the complexity of interactions involved and the co-variability between individual components, the magnitude and sign of these feedbacks remain uncertain.

This study uses a multitude of datasets to characterize the $N_d$ seasonal cycle and factors related to $N_d$ variability. The structure of the results and discussion is as follows: (i) case study flight highlighting the wide range of $N_d$ in wintertime and factors potentially affecting that variability; (ii) seasonal cycle of $N_d$ and aerosol concentrations based on different proxy variables; (iii) seasonal cycles of factors potentially influential for $N_d$ such as aerosol size distribution, vertical distribution of aerosol, humidity effects, and aerosol–cloud interactions; (iv) composite analysis of influential factors on high- and low-$N_d$ days in each season; (v) modeling analysis to probe more deeply into $N_d$ relationships with other parameters for winter and summer seasons; and (vi) discussion of other factors relevant to $N_d$ unexplored in this work.

## 2   Methods

### 2.1   Study region

We focus on the WNAO, defined here as being bounded by 25–50° N and 60–85° W. A subset of the results focuses on six individual sub-domains representative of different parts of the WNAO (shown later), with five just off the East Coast extending from south to north (south: S; central-south: C-S; central: C; central-north: C-N; north: N) and one over Bermuda.

### 2.2   Datasets

#### 2.2.1   Satellite observations (CERES-MODIS/CALIPSO)

Relevant cloud parameters were obtained from the Clouds and the Earth's Radiant Energy System (CERES) edition 4 products (Minnis et al., 2011, 2020), which are based on the application of CERES's retrieval algorithms on the radiances measured by the MODerate resolution Imaging Spectroradiometer (MODIS) instrument aboard the Aqua satellite. Aqua observations used to estimate $N_d$ were from the daytime overpasses of the satellite around 13:30 local time (LT). Level 3 daily cloud properties at 1° × 1° spatial resolution (listed in Table 1) were used for the period between January 2013 and December 2017 from CERES-MODIS edition 4 Single Scanning Footprint (SSF) products (Loeb et al., 2016). The CERES-MODIS SSF Level 3 product includes 1° × 1° averaged data according to the cloud-top pressure of individual pixels: low (heights below 700 hPa), mid-low

**Atmos. Chem. Phys., 21, 1–28, 2021**                    **https://doi.org/10.5194/acp-21-1-2021**

(heights within 700–500 hPa), mid-high (heights within 500–300 hPa), and high (heights above 300 hPa) level clouds. For this study, we only use low-cloud averages.

$N_d$ is estimated based on an adiabatic cloud model (Grosvenor et al., 2018):

$$N_d = \frac{\sqrt{5}}{2\pi k}\left(\frac{f_{ad}C_w\tau}{Q_{ext}\rho_w r_e^5}\right)^{1/2}, \qquad (1)$$

where $\tau$ is cloud optical depth and $r_e$ is cloud drop effective radius, both of which are obtained from CERES-MODIS for low-level (i.e., surface to 700 hPa) liquid clouds. $Q_{ext}$ is the unitless extinction efficiency factor, assumed to be 2 for liquid cloud droplets, and $\rho_w$ is the density of water (1 g cm$^{-3}$). Methods described in Painemal (2018) were used to estimate parameters in Eq. (1) as follows. (i) Adiabatic water lapse rate ($C_w$) was determined using cloud-top pressure and temperature provided by CERES-MODIS. (ii) The $N_d$ estimation is often corrected for the sub-adiabatic profile by applying the adiabatic value ($f_{ad}$), but in this work, a value of $f_{ad} = 1$ was assumed due to both lack of consensus on its value and its relatively minor impact on $N_d$ estimation (Grosvenor et al., 2018). (iii) $k$ is the parameter representing the width of the droplet spectrum and was assumed to be 0.8 over the ocean. Statistics of $N_d$ are often estimated after screening daily observations based on cloud fractions (Wood, 2012; Grosvenor et al., 2018). The purpose of such filters is to reduce the uncertainties associated with the estimation of $N_d$ (Eq. 1) driven by the errors in the retrieval of $r_e$ and $\tau$ from MODIS's observed reflectance in a highly heterogeneous cloud field. However, this may inadvertently mask the effects of cloud regime on aerosol–cloud interactions by only including certain low-level cloud types in the analyses (e.g., closed-cell stratocumulus). Therefore, we use all $N_d$ data regardless of cloud fraction with exceptions being Sects. 3.5 and 4.2 where a filter of low-level liquid cloud fraction (i.e., CF$_{low-liq.} \geq 0.1$) was applied.

The Cloud-Aerosol Lidar with Orthogonal Polarization (CALIOP) instrument aboard the Cloud-Aerosol Lidar and Infrared Pathfinder Satellite Observations (CALIPSO) provides data on the vertical distribution of aerosols (Winker et al., 2009). Nighttime extinction profiles were acquired from Level 2 version 4.20 products (i.e., 5 km aerosol profile data), between January 2013 and December 2017. We averaged the Level 2 daily extinctions in different $4° \times 5°$ sub-domains (shown later) to obtain the seasonal profiles after applying the screening scheme outlined in Tackett et al. (2018).

### 2.2.2 MERRA-2

Aerosol data were obtained from the Modern-Era Retrospective Analysis for Research and Applications-Version 2 (MERRA-2) (Gelaro et al., 2017). MERRA-2 is a multidecadal reanalysis where meteorological and aerosol observations are jointly assimilated into the Goddard Earth Observation System version 5 (GEOS-5) data assimilation system (Buchard et al., 2017; Randles et al., 2017). Aerosols in MERRA-2 are simulated with a radiatively coupled version of the Goddard Chemistry, Aerosol, Radiation, and Transport model (GOCART; Chin et al., 2002; Colarco et al., 2010). GOCART treats the sources, sinks, and chemistry of 15 externally mixed aerosol mass mixing ratio tracers, which include sulfate, hydrophobic and hydrophilic black and organic carbon, dust (five size bins), and sea salt (five size bins). MERRA-2 includes assimilation of bias-corrected Collection 5 MODIS AOD, bias-corrected AOD from the Advanced Very High Resolution Radiometer (AVHRR) instruments, AOD retrievals from the Multiangle Imaging Spectro-Radiometer (MISR) over bright surfaces, and ground-based Aerosol Robotic Network (AERONET) direct measurements of AOD (Gelaro et al., 2017). In this study we used total and speciated (i.e., sea salt, dust, black carbon, organic carbon, and sulfate) AOD at 550 nm between January 2013 and December 2017 at times relevant to Aqua's overpass time (13:30 LT). Aerosol index was calculated as the product of AOD and the Ångström parameter. MERRA-2 also provides surface mass concentrations of aerosol species including sea salt, dust, black carbon, organic carbon, and sulfate, which were used as a measure of aerosol levels in the planetary boundary layer (PBL).

MERRA-2 data were also used for environmental variables including both thermodynamic (e.g., temperature and relative humidity) and dynamic parameters (e.g., sea level pressure (SLP) and geopotential heights) (Gelaro et al., 2017) listed in Table 1. Bilinear interpolation was applied to transfer all MERRA-2 variables (Table 1) from their original $0.5° \times 0.625°$ spatial resolution to the equivalent $1° \times 1°$ grid in CERES-MODIS Level 3 data.

### 2.2.3 Precipitation data

Daily precipitation data were obtained from the Precipitation Estimation from Remotely Sensed Information using Artificial Neural Networks–Climate Data Record (PERSIANN-CDR) data product (Ashouri et al., 2015; Nguyen et al., 2018). Bilinear interpolation was applied to convert the PERSIANN-CDR data from their native spatial resolution (i.e., $0.25° \times 0.25°$) to equivalent $1° \times 1°$ grids in CERES-MODIS Level 3 data. It is important to note that we use daily averaged PERSIANN-CDR precipitation and, therefore, there is some temporal mismatch with the daily $N_d$ value from MODIS-Aqua that comes at one time of the day. This can contribute to some level of uncertainty for the discussions based on analyses involving relationships between precipitation and $N_d$.

### 2.2.4 Surface-based CCN data

Cloud condensation nuclei (CCN) data were obtained from the U.S. Department of Energy's Two-Column Aerosol Project (TCAP) (Berg et al., 2016) to examine the seasonal

**Table 1.** Summary of various data products used in this study. TS2

| Parameter | Data source | Spatial resolution | Vertical level | Date range | Spatial area | Temporal resolution |
|---|---|---|---|---|---|---|
| Cloud optical thickness | CERES-MODIS | 1° × 1° | NA | 1 January 2013–31 December 2017 | 25–50° N, 60–85° W | Daily |
| Cloud effective radius | CERES-MODIS | 1° × 1° | NA | 1 January 2013–31 December 2017 | 25–50° N, 60–85° W | Daily |
| Cloud fraction | CERES-MODIS | 1° × 1° | NA | 1 January 2013–31 December 2017 | 25–50° N, 60–85° W | Daily |
| Cloud-top temperature | CERES-MODIS | 1° × 1° | NA | 1 January 2013–31 December 2017 | 25–50° N, 60–85° W | Daily |
| Cloud effective height | CERES-MODIS | 1° × 1° | NA | 1 January 2013–31 December 2017 | 25–50° N, 60–85° W | Daily |
| Cloud-top pressure | CERES-MODIS | 1° × 1° | NA | 1 January 2013–31 December 2017 | 25–50° N, 60–85° W | Daily |
| Precipitation | PERSIANN-CDR | 1° × 1° | NA | 1 January 2013–31 December 2017 | 25–50° N, 60–85° W | Daily |
| Aerosol extinction (532 nm) | CALIPSO/CALIOP | 5 km | NA | 1 January 2013–31 December 2017 | 25–50° N, 60–85° W | Daily |
| Total aerosol extinction AOT (550 nm) | MERRA-2 | 1° × 1° | NA | 1 January 2013–31 December 2017 | 25–50° N, 60–85° W | Daily |
| Total aerosol Ångström parameter (470–870 nm) | MERRA-2 | 1° × 1° | NA | 1 January 2013–31 December 2017 | 25–50° N, 60–85° W | Daily |
| Sulfate extinction AOT (550 nm) | MERRA-2 | 1° × 1° | NA | 1 January 2013–31 December 2017 | 25–50° N, 60–85° W | Daily |
| Sea salt extinction AOT (550 nm) | MERRA-2 | 1° × 1° | NA | 1 January 2013–31 December 2017 | 25–50° N, 60–85° W | Daily |
| Dust extinction AOT (550 nm) | MERRA-2 | 1° × 1° | NA | 1 January 2013–31 December 2017 | 25–50° N, 60–85° W | Daily |
| Organic carbon extinction AOT (550 nm) | MERRA-2 | 1° × 1° | NA | 1 January 2013–31 December 2017 | 25–50° N, 60–85° W | Daily |
| Black carbon extinction AOT (550 nm) | MERRA-2 | 1° × 1° | NA | 1 January 2013–31 December 2017 | 25–50° N, 60–85° W | Daily |
| Sulfate surface mass concentration | MERRA-2 | 1° × 1° | NA | 1 January 2013–31 December 2017 | 25–50° N, 60–85° W | Daily |
| Sea salt surface mass concentration | MERRA-2 | 1° × 1° | NA | 1 January 2013–31 December 2017 | 25–50° N, 60–85° W | Daily |
| Dust surface mass concentration | MERRA-2 | 1° × 1° | NA | 1 January 2013–31 December 2017 | 25–50° N, 60–85° W | Daily |
| Organic carbon surface mass concentration | MERRA-2 | 1° × 1° | NA | 1 January 2013–31 December 2017 | 25–50° N, 60–85° W | Daily |
| Black carbon surface mass concentration | MERRA-2 | 1° × 1° | NA | 1 January 2013–31 December 2017 | 25–50° N, 60–85° W | Daily |
| Sea level pressure | MERRA-2 | 1° × 1° | Surface | 1 January 2013–31 December 2017 | 25–50° N, 60–85° W | Daily |
| Geopotential height | MERRA-2 | 1° × 1° | 850 hPa | 1 January 2013–31 December 2017 | 25–50° N, 60–85° W | Daily |
| Sea surface temperature | MERRA-2 | 1° × 1° | Sea surface | 1 January 2013–31 December 2017 | 25–50° N, 60–85° W | Daily |
| Air temperature | MERRA-2 | 1° × 1° | Surface, 850, 700 hPa | 1 January 2013–31 December 2017 | 25–50° N, 60–85° W | Daily |
| Relative humidity | MERRA-2 | 1° × 1° | 1000–500 hPa | 1 January 2013–31 December 2017 | 25–50° N, 60–85° W | Daily |
| Wind speed | MERRA-2 | 1° × 1° | 2 m, 950 hPa | 1 January 2013–31 December 2017 | 25–50° N, 60–85° W | Daily |
| Planetary boundary layer height | MERRA-2 | 1° × 1° | NA | 1 January 2013–31 December 2017 | 25–50° N, 60–85° W | Daily |
| Vertical pressure velocity | MERRA-2 | 1° × 1° | 800 hPa | 1 January 2013–31 December 2017 | 25–50° N, 60–85° W | Daily |
| Aerosol/cloud | Airborne: ACTIVATE | — | NA | 22 February 2020 | 34.08–37.16° N, 72.31–76.64° W | 1 s |
| CCN-Cape Cod | Ground-based measurement | Point measurement | Surface | 16 July 2012–4 May 2013 | 41.67° N, 70.30° W | 1 s |

variations in CCN number concentration at a representative site by Cape Cod, Massachusetts (41.67° N, 70.30° W), over the US East Coast. TCAP was a campaign conducted between June 2012 and June 2013 to investigate aerosol optical and physicochemical properties and interactions between aerosols and clouds (Berg et al., 2016; Liu and Li, 2019). CCN data were available between July 2012 and May 2013 at multiple supersaturations with some gaps in the data collection (i.e., November–December); for simplicity, we focused on CCN data measured at a single supersaturation of 1 % owing to relatively better data coverage compared to lower supersaturations. We note that this higher supersaturation is not necessarily representative of that relevant to the clouds of interest but is still insightful for understanding the seasonal cycle of CCN concentration. The qualitative seasonal cycle of CCN concentration at 1 % matches those at lower supersaturations (e.g., 0.15 %–0.8 %).

### 2.2.5 Airborne in situ data

We used airborne in situ data collected during the fifth research flight (RF05) of the Aerosol Cloud meTeorology Interactions oVer the western ATlantic Experiment (ACTI-VATE) campaign. One flight is used both for simplicity and because it embodied conditions relevant to the discussion of other results. The mission concept involves joint flights between the NASA Langley UC-12 King Air and HU-25 Falcon such that the former flies around 8–10 km, and the latter flies in the boundary layer to simultaneously collect data on aerosol, cloud, gas, and meteorological parameters in the same column (Sorooshian et al., 2019). The Falcon flew in a systematic way to collect data at different vertical regions relative to cloud, including the following of relevance to this study: BCB – below cloud base; ACB – above cloud base; BCT – below cloud top; and Min. Alt – minimum altitude the plane flies at ($\sim 150$ m).

This study makes use of the HU-25 Falcon data from the following instruments: fast cloud droplet probe (FCDP; $D_p \sim 3$–$50$ µm) (SPEC Inc.) aerosol and cloud droplet size distributions for quantification of cloud liquid water content (LWC), $N_d$, and aerosol number concentrations with $D_p$ exceeding 3 µm in cloud-free air (termed FCDP-aerosol); two-dimensional stereo (2DS; $D_p \sim 28.5$–$1464.9$ µm) (SPEC Inc.) probe for estimation of rain water content (RWC) by integrating raindrop ($D_p \geq 39.9$ µm) size distributions; cloud condensation nuclei (CCN; DMT) counter for CCN number concentrations; laser aerosol spectrometer (LAS; TSI model 3340) and condensation particle counter (CPC; TSI model 3772) for aerosol number concentrations with $D_p$ between 0.1–1 µm and above 10 nm, respectively; high-resolution time-of-flight aerosol mass spectrometer (AMS; Aerodyne) for submicrometer non-refractory aerosol composition (DeCarlo et al., 2008), operated in 1 Hz Fast-MS mode and averaged to 25 s time resolution; and turbulent air-motion

measurement system (TAMMS) for winds and temperature (Thornhill et al., 2003).

CCN, LAS, CPC, and AMS data were collected downstream of an isokinetic double diffuser inlet (BMI, Inc.), whereas the AMS and LAS also sampled downstream of a counterflow virtual impactor (CVI) inlet (BMI, Inc.) when in cloud (Shingler et al., 2012). However, a filter was applied to remove LAS data when the CVI inlet was used. Measurements from the CCN counter, LAS, CPC, and FCDP-aerosol are only shown in cloud-free and rain-free conditions, distinguished by LWC $< 0.05$ g m$^{-3}$ and RWC $< 0.05$ g m$^{-3}$, respectively, and also excluding data collected 20 s before and after evidence of rain or cloud. Estimation of supermicrometer particles from FCDP measurements was performed after conducting the following additional screening steps to minimize cloud droplet artifacts: (i) only samples with RH $< 98$ % were included; (ii) data collected during ACB and BCT legs were excluded. CCN, LAS, CPC, and AMS measurements are reported at standard temperature and pressure (i.e., 273 K and 101.325 kPa) while FCDP and 2DS measurements correspond to ambient conditions.

### 2.3 Regression analyses

Regression modeling was conducted to investigate relationships between environmental variables and $N_d$. The gradient-boosted regression trees (GBRT) model, classified as a machine learning (ML) model, is used, consisting of several weak learners (i.e., regression trees with a fixed size) that are designed and subsequently trained to improve prediction accuracy by fitting the model's trees on residuals rather than response values (Hastie et al., 2009). Desirable characteristics of the GBRT model include both its capacity to capture nonlinear relationships and being less vulnerable to overfitting (Persson et al., 2017; Fuchs et al., 2018; Dadashazar et al., 2020). Two separate GBRT models were trained using daily CERES-MODIS $N_d$ data (1° × 1°) in winter (DJF) and summer (JJA) to reveal potential variables impacting $N_d$. Winter and summer are chosen as they exhibit the highest and lowest $N_d$ concentrations, respectively, among all seasons over the WNAO.

Many variables were picked as input parameters (Table 2) for the GBRT model, categorized as being aerosol, dynamic/thermodynamic, or cloud variables. Aerosol parameters included MERRA-2 surface mass concentrations for sulfate, sea salt, dust, and organic carbon. Black carbon concentration was removed from input parameters because of its high correlation ($R^2 = 0.6$) with organic carbon. The following is the list of thermodynamic/dynamic input parameters derived from MERRA-2: vertical pressure velocity at 800 hPa ($\omega_{800}$), planetary boundary layer height (PBLH), cold-air outbreak (CAO) index, wind speed and wind direction at 2 m (wind$_{2m}$ and wind-dir$_{2m}$), and relative humidity (RH) in the PBL and free troposphere represented by RH$_{950}$ and RH$_{800}$, respectively. CAO index is defined as the dif-

**Table 2.** List of input parameters used as predictor variables in the GBRT and linear models. Variables are grouped into three general categories.

| | Parameter |
|---|---|
| Aerosol | Sulfate surface mass concentration (Sulfate$_{sf-mass}$) |
| | Sea salt surface mass concentration (Sea-salt$_{sf-mass}$) |
| | Dust surface mass concentration (Dust$_{sf-mass}$) |
| | Organic carbon surface mass concentration (OC$_{sf-mass}$) |
| Cloud | Low-level liquid cloud fraction (CF$_{low-liq.}$) |
| | Low-level liquid cloud-top effective height (Cloud-top$_{low-liq.}$) |
| | Precipitation rate (Rain) |
| Dynamic/ thermodynamic | Cold-air outbreak index (CAO$_{index}$): $\theta^*_{skt}$-$\theta_{850}$ |
| | Relative humidity at 950 hPa (RH$_{950}$) |
| | Relative humidity at 800 hPa (RH$_{800}$) |
| | Vertical pressure velocity at 800 hPa ($\omega_{800}$) |
| | Wind speed at 2 m (Wind$_{2\,m}$) |
| | Wind direction at 2 m (Wind-dir$_{2\,m}$) |
| | Planetary boundary layer height (PBLH) |

* Skin potential temperature.

ference between skin potential temperature ($\theta_{skt}$) and air potential temperature at 850 hPa ($\theta_{850}$) (Papritz et al., 2015). Updraft velocity plays a crucial role in the activation of aerosol into cloud droplets in warm clouds (Feingold, 2003; Reutter et al., 2009). Since the direct representation of updraft speed is not available from reanalysis data, near-surface wind speed (i.e., wind$_{2\,m}$) is used as a representative proxy parameter as an input parameter to the regression models. CERES-MODIS cloud parameters include liquid cloud fraction and cloud-top height for low-level clouds. In addition, PERSIANN-CDR daily precipitation (Rain) was included as a relevant cloud parameter.

Data were split into two sets: training/validation (70 %) and testing (30 %). Five-fold cross-validation was implemented to train the GBRT model using the training/validation data. Furthermore, both performance and generalizability of the trained models were tested via the aid of the test set, which was not used in the training process. Hyperparameters of the GBRT models were optimized through a combination of both random and grid search methods. Table S1 in the Supplement shows the list of important hyperparameters of the GBRT model and associated ranges tested via random and grid search methods. The optimized model hyperparameters can also be found in Table S1. The GBRT models were performed using the scikit-learn module designed in Python (Pedregosa et al., 2011).

The regression analyses were not performed solely to construct and provide a highly accurate model useful for prediction, but rather to disclose and examine the possible effects of the relevant input variables on $N_d$ considering all the shortcomings of such analyses. For instance, there is some level of interdependency between input variables. To reduce unwanted consequences of correlated features, the interpre-

tation of the results was done with the aid of accumulated local effect (ALE) plots, which are specifically designed to be unbiased to the correlated input variables (Apley and Zhu, 2020). ALE plots illustrate the influence of input variables on the response parameter in ML models. The ALE value for a particular variable $s$ at a specific value of $x_s$ (i.e., $f_{s,ALE}(x_s)$) can be calculated as follows:

$$f_{s,ALE}(x_s) = \int_{z_{0,1}}^{x_s} \int_{x_c} f^s(z_s, x_c) P(x_c|z_s) \,dx_c dz_s - \text{constant}, \quad (2)$$

where $f^s(z_s x_c)$ is the gradient of model's response with respect to variable $s$ (i.e., local effect) and $P(x_c|z_s)$ is the conditional distribution of $x_c$, where $c$ denotes the other input variables rather than $s$, and $x_c$ is the associated point in the variable space of $c$. $z_{0,1}$ is chosen arbitrarily below the smallest observation of feature $s$ (Apley and Zhu, 2020). The steps in Eq. (2) can be summarized as follows (Molnar, 2019; Apley and Zhu, 2020): (i) the average change in the model's prediction is calculated using the conditional distribution of features; (ii) the average change will then be accumulated by integrating it over feature $s$; and (iii) a constant will be subtracted to vertically center (i.e., the average of ALE becomes zero) the ALE plot. The aforementioned steps, although seemingly complex, assure the avoidance of undesired extrapolation (especially an issue for correlated variables) occurring in alternative approaches such as partial dependence (PD) plots. The value of $f_{s,ALE}(x_s)$ can be viewed as the difference between the model's response at $x_s$ and the average prediction. We used the source code available in https://github.com/blent-ai/ALEPython (Jumelle et al., 2021) for the calculation of ALE plots.

## 3 Results and discussion

### 3.1 Aircraft case study of $N_d$ gradient

ACTIVATE Research Flight 5 (RF05) on 22 February 2020 demonstrates the wide range in $N_d$ offshore in the PBL (1.6 km) over the WNAO (Fig. 1). On this day, the ACTIVATE study region was dominated by a surface high-pressure system centered over the southeastern US, with a significant ridge axis extending from the main high to the east-northeast off the Virginia–North Carolina coast and into the WNAO. Aloft, the flight region was located in northwesterly flow behind a trough offshore. This setup led to subsidence in the region and generally clear skies, except where scattered to broken marine boundary layer clouds formed along and east of the Gulf Stream. The 2 d NOAA HYSPLIT (Stein et al., 2015; Rolph et al., 2017) back trajectories using the "model vertical velocity" method and "REANALYSIS" meteorology data indicate air in the flight region (between 0–3 km) had wrapped around the surface high from the north and left the New England coast 12–24 h beforehand (with a descending profile). Along the flight segment shown, winds were approximately 6 m s$^{-1}$, out of the northnorthwest during the initial descent, Min. Alt. 1, and BCB1 legs and primarily from the northeast for the other sections of the flight. Sea surface temperatures were 6–9 °C near the coast during the descent and Min. Alt. 1 leg (readers are referred to Fig. 1's caption for the definition of different legs); 21–25 °C over the Gulf Stream during the BCB1, ACB1, and BCB2 legs; and 17–20 °C for the remainder of the flight segment shown. The majority of the segment was in or below the boundary layer clouds, with cloud base around 900–1100 m and cloud top around 1750 m. Note that the initial BCB1 leg was much lower at around 460 m, likely reflecting a shallower marine boundary layer and cloud base near the much colder waters close to the coast. Static air temperature ranged between 0–10 °C, except for the BCT1 leg where temperatures were around −2.3 °C.

$N_d$ values from the FCDP ranged from a maximum value of 1298 cm$^{-3}$ closer to the coast during the ACB1 leg (35.00° N, 74.55° W) to a minimum of 19 cm$^{-3}$ farther away in the BCT1 leg (34.32° N, 72.73° W). The minimum $N_d$ value in the ACB3 leg was 85 cm$^{-3}$ (34.11° N, 72.80° W), which is a fairer comparison to the ACB1 leg compared to the BCT1 leg in terms of being closer to cloud base. The mean $N_d$ values (cm$^{-3}$) in the cloudy portions of the ACB1, BCT1, and ACB3 legs were as follows: 849, 77, and 143.

Based on the nearest BCB legs adjacent to the maximum and minimum $N_d$ values (BCB1 = 35.31° N, 74.95° W; BCB3 = 34.41° N, 72.70° W), there was a significant offshore gradient in LAS submicrometer particle number concentration and AMS non-refractory aerosol mass, ranging from as high as 424 cm$^{-3}$ and 5.60 µg m$^{-3}$ (during BCB1) to as low as 21 cm$^{-3}$ and 0.27 µg m$^{-3}$ (during BCB3). The mean values of submicrometer particle number concentration and AMS non-refractory aerosol for the two BCB legs were as follows: 277 cm$^{-3}$/3.64 µg m$^{-3}$ (BCB1) and 48 cm$^{-3}$/0.42 µg m$^{-3}$ (BCB3). The higher $N_d$ value (1298 cm$^{-3}$) relative to LAS aerosol concentration (424 cm$^{-3}$) at the near-shore point is suggestive of aerosol smaller than 0.1 µm activating into drops. This is supported by the fact that both CCN (supersaturation = 0.43 %) and CPC number concentrations with $D_p > 10$ nm exhibited mean values of 980 and 1723 cm$^{-3}$ in the BCB1 leg, respectively, dropping to 98 and 260 cm$^{-3}$ in the BCB3 leg. For the duration of the flight portion shown in Fig. 1, supermicrometer concentrations varied over 2 orders of magnitude (0.002–0.51 cm$^{-3}$) and expectedly did not exhibit a pronounced offshore gradient as it is naturally emitted from the ocean.

Closer to shore during the Min. Alt. 1 leg, nitrate was the dominant aerosol species (∼ 70 % mass fraction). Farther offshore during both the BCB1 leg and cloud-free portion of the ACB1 leg, organics were the dominant constituent (∼ 46 % mass fraction), whereas farther during the BCB3 leg, the mean mass fraction of sulfate was the highest (75 %). Droplet residual particle data show a greater contribution of organics farther offshore, increasing from 46 % to 75 % between the ACB1 and ACB3 legs, respectively. These composition results, albeit limited to the non-refractory portion of submicrometer aerosol particles, reveal significant changes with distance offshore indicative of varying chemical properties of particles activating into droplets.

The cloudy portions of ACB1 are characterized as having little or no rain with a maximum RWC value of 0.02 g m$^{-3}$ and mean value of 0.003 g m$^{-3}$. There is a notable RWC peak at the beginning of the Min. Alt. 2 leg, reaching as high as 1.81 g m$^{-3}$ associated with clouds aloft. The precipitation occurrence was also evident in a subsequent BCT1 leg where RWC reached as high as 0.18 g m$^{-3}$. GOES satellite imagery of the study region (Fig. 1) also reflects the effect of precipitation on cloud morphology where clouds farther offshore resemble open-cell structures. Associated scavenging of particles through the washout process is presumed to contribute to the decline in aerosol concentrations with distance offshore.

Figure 1 shows changes in aerosol characteristics coincident with the large gradient in $N_d$. While ACTIVATE airborne data collection is ongoing to build flight statistics over multiple years, the wide changes in microphysical properties in RF05 motivate looking at other datasets with broader spatiotemporal coverage to learn about potential seasonally dependent drivers of $N_d$, including meteorological parameters that vary throughout the year. Furthermore, other datasets can provide insight into the source(s) of seasonal discrepancy between columnar aerosol remote sensing parameters and $N_d$.

### 3.2 Seasonal cycles of $N_d$ and AOD

Figure 2 illustrates the seasonal differences in MERRA-2 AOD and CERES-MODIS $N_d$ over the WNAO that partly motivate this study. Seasonal mean values (±

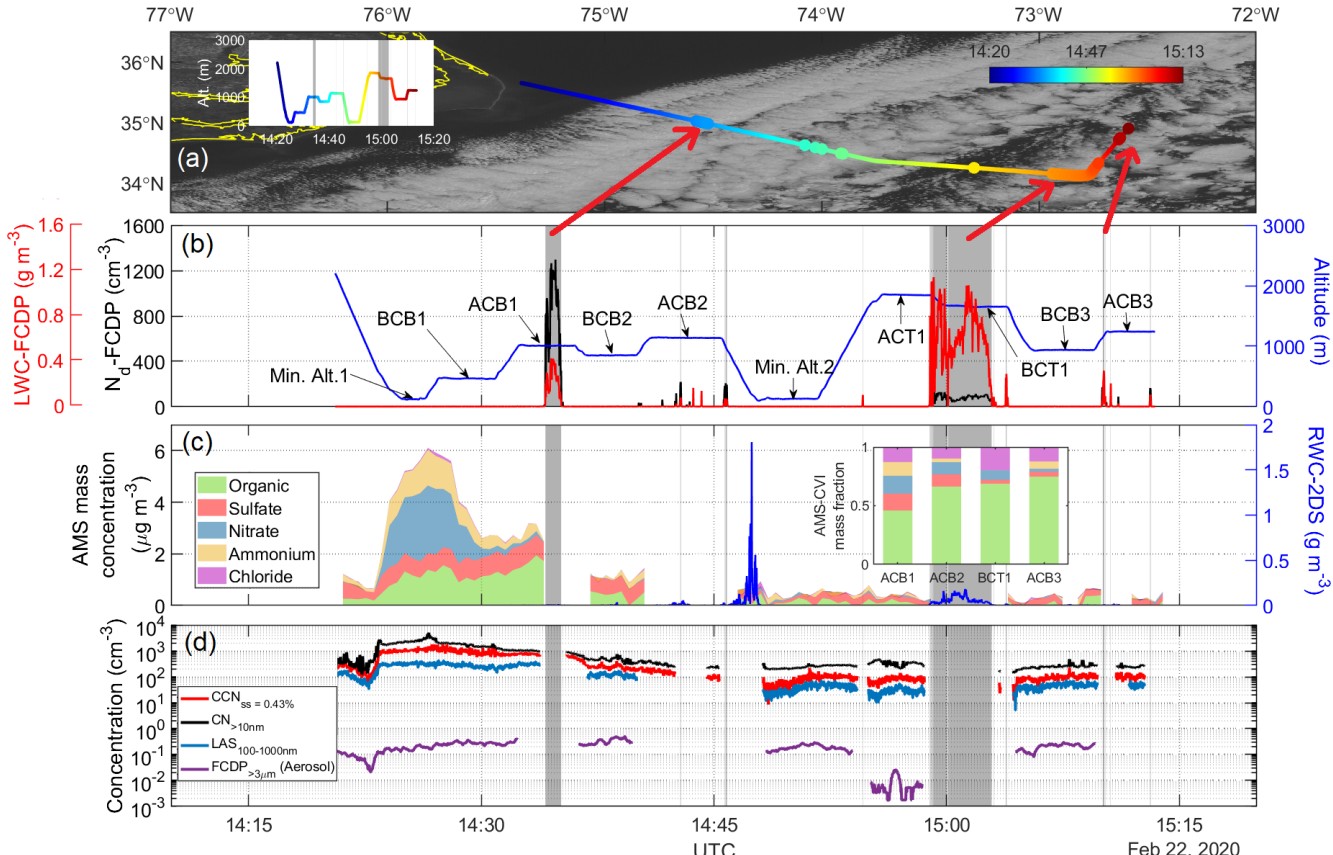

**Figure 1.** Time series of selected parameters measured by the HU-25 Falcon aircraft during a selected segment of RF05 on 22 February 2020. **(a)** Overlayed flight track on GOES 16 visible imagery obtained at 14:55:04 UTC. TS3 **(b)** Altitude, cloud liquid water content (LWC), and $N_d$, with the latter two obtained from the FCDP. **(c)** Rain water content (RWC) measured by 2DS probe, AMS speciated mass concentration in cloud/rain-free air, and AMS mass fractions for droplet residual particles in cloud as measured downstream of a CVI inlet. **(d)** Number concentrations for CCN at 0.43 % supersaturation and particles for three diameter ranges: above 10 nm (CPC), 100–1000 nm (LAS), and above 3 μm (FCDP). Shaded gray areas in panels **(b)**–**(d)** highlight cloudy periods identified as having LWC $\geq 0.05\,\mathrm{g\,m^{-3}}$. Locations of the cloudy regions are pointed to with red arrows in the satellite imagery. Level legs are defined as follows: BCB: below cloud base; ACB: above cloud base; Min. Alt.: minimum altitude the plane flies at ($\sim$ 150 m); ACT: above cloud top; BCT: below cloud top.

standard deviation) of AOD/$N_d$ (cm$^{-3}$) were as follows for the entire WNAO: DJF $= 0.11 \pm 0.03 / 64.1 \pm 18.0$; MAM $= 0.16 \pm 0.03 / 60.4 \pm 13.1$; JJA $= 0.15 \pm 0.03 / 49.1 \pm 10.1$; and SON $= 0.11 \pm 0.03 / 50.3 \pm 13.9$. In contrast to AOD, $N_d$ values and low-cloud fraction (Fig. 2c) were highest in DJF and lowest in JJA. DJF showed notably high $N_d$ near the coast, qualitatively consistent with the airborne data. The seasons with the greatest AOD values, accompanied by the most pronounced spatial gradient offshore, were JJA and MAM. The offshore gradient owes to continental pollution outflow (Corral et al., 2021, and references therein). In contrast, DJF and SON exhibited lower AOD values with a distinct area of higher AOD values offshore between $\sim$ 35–40° N accounted for by sea salt. MERRA-2 speciated AOD data (Fig. 3) indicate that sea salt and sulfate dominate total AOD regardless of season and that sulfate, organic carbon, and black carbon most closely follow the offshore gradient

pattern owing to continental sources. Dust and sea salt have different spatial distributions with the former derived from sources such as North Africa leading to enhanced AODs $< 30°$ N in particular in JJA and sea salt being enhanced offshore in particular in JJA.

Table 3 probes deeper into individual WNAO sub-domains to compare seasonal AOD and $N_d$ values. For the six sub-domains in Fig. 2, MERRA-2 AOD peaks in MAM and JJA, while $N_d$ peaks in DJF. The Bermuda sub-domain was unique in that mean $N_d$ was slightly higher in MAM (53 cm$^{-3}$) compared to DJF (48 cm$^{-3}$). We attribute the slightly different seasonal cycle over Bermuda to its remote nature, leading to differences in meteorology and aerosol sources between seasons.

One factor that could bias AOD towards higher values with disproportionately less impact on $N_d$ is aerosol hygroscopic growth in humid conditions. Table 3 summarizes mean MERRA-2 RH values in the PBL and free troposphere

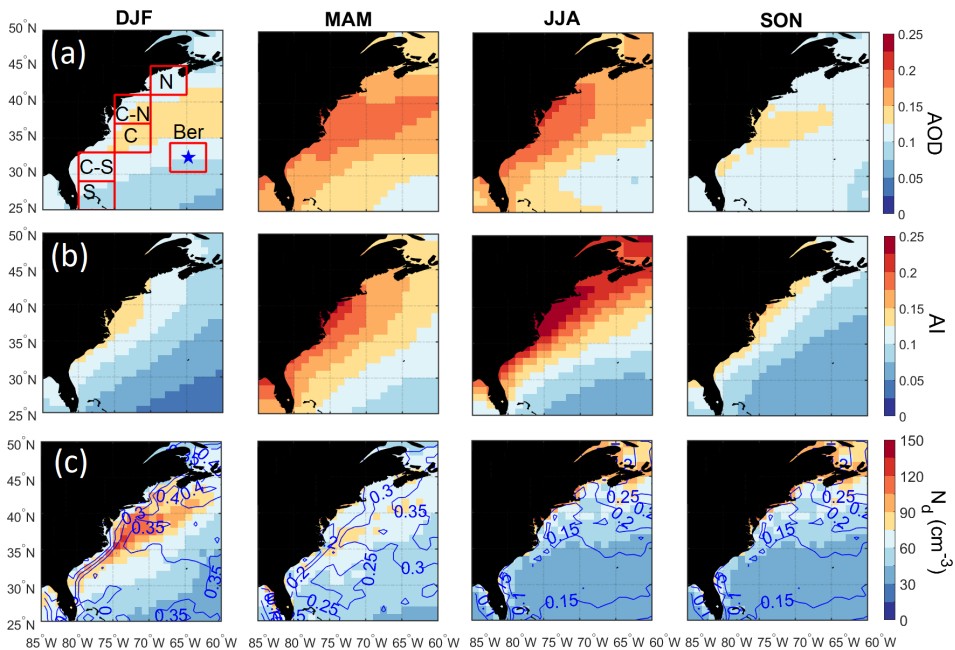

**Figure 2.** Seasonal spatial maps for **(a)** MERRA-2 aerosol optical depth (AOD), **(b)** MERRA-2 aerosol index (AI), and **(c)** cloud drop number concentration ($N_d$) over the western North Atlantic Ocean (WNAO). Contours in panel **(c)** represent low-level (cloud-top pressure > 700 hPa) liquid cloud fraction ($CF_{low-liq.}$). Cloud data are based on daily Level 3 data from CERES-MODIS. The maps are based on data between January 2013 and December 2017. The boxes in the top left panel represent sub-domains examined in more detail throughout the study, with the blue star denoting Bermuda.

**Table 3.** Average drop number concentration ($N_d$), MERRA-2 AOD, and vertically resolved AOD characteristics from CALIOP for each season over the sub-domains shown in Fig. 2. Total CALIOP AOD is shown outside parentheses, and numbers inside are the percent AOD fraction in the planetary boundary layer followed by in the free troposphere. Also shown are PBLHs (shown in Fig. 4) and the relative humidity in the PBL and FT. TS4

|  | S | C-S | C | C-N | N | Bermuda |
|---|---|---|---|---|---|---|
| $AOD_{MERRA-2}/N_d$ (cm$^{-3}$) | | | | | | |
| DJF | 0.10/56 | 0.11/74 | 0.13/91 | 0.12/97 | 0.11/78 | 0.10/48 |
| MAM | 0.14/55 | 0.17/62 | 0.18/72 | 0.19/75 | 0.16/70 | 0.14/53 |
| JJA | 0.14/41 | 0.16/43 | 0.17/47 | 0.19/68 | 0.17/73 | 0.11/37 |
| SON | 0.11/42 | 0.12/53 | 0.13/62 | 0.13/74 | 0.11/73 | 0.11/36 |
| $AOD_{CALIOP}$ (%PBL, %FT) | | | | | | |
| DJF | 0.11 (64,36) | 0.11 (67,33) | 0.15 (68,32) | 0.09 (61,39) | 0.13 (59,41) | 0.14 (72,28) |
| MAM | 0.11 (54,46) | 0.10 (53,47) | 0.12 (58,42) | 0.10 (30,70) | 0.07 (30,70) | 0.12 (58,42) |
| JJA | 0.11 (53,47) | 0.11 (44,56) | 0.10 (46,54) | 0.11 (20,80) | 0.08 (11,89) | 0.08 (49,51) |
| SON | 0.09 (63,37) | 0.10 (57,43) | 0.10 (65,35) | 0.08 (47,53) | 0.07 (35,65) | 0.10 (69,31) |
| PBLH (m)/$RH_{PBL}$ (%)/$RH_{FT}$ (%) | | | | | | |
| DJF | 1018/78/37 | 1156/76/43 | 1364/79/46 | 1013/76/52 | 926/76/58 | 1198/80/43 |
| MAM | 903/77/41 | 955/72/43 | 1043/75/48 | 722/72/53 | 568/79/55 | 966/79/50 |
| JJA | 775/81/62 | 725/81/60 | 697/81/59 | 481/78/53 | 351/85/55 | 713/82/58 |
| SON | 1018/80/50 | 1094/76/45 | 1181/76/42 | 825/71/43 | 593/77/51 | 1095/81/48 |

**https://doi.org/10.5194/acp-21-1-2021** **Atmos. Chem. Phys., 21, 1–28, 2021**

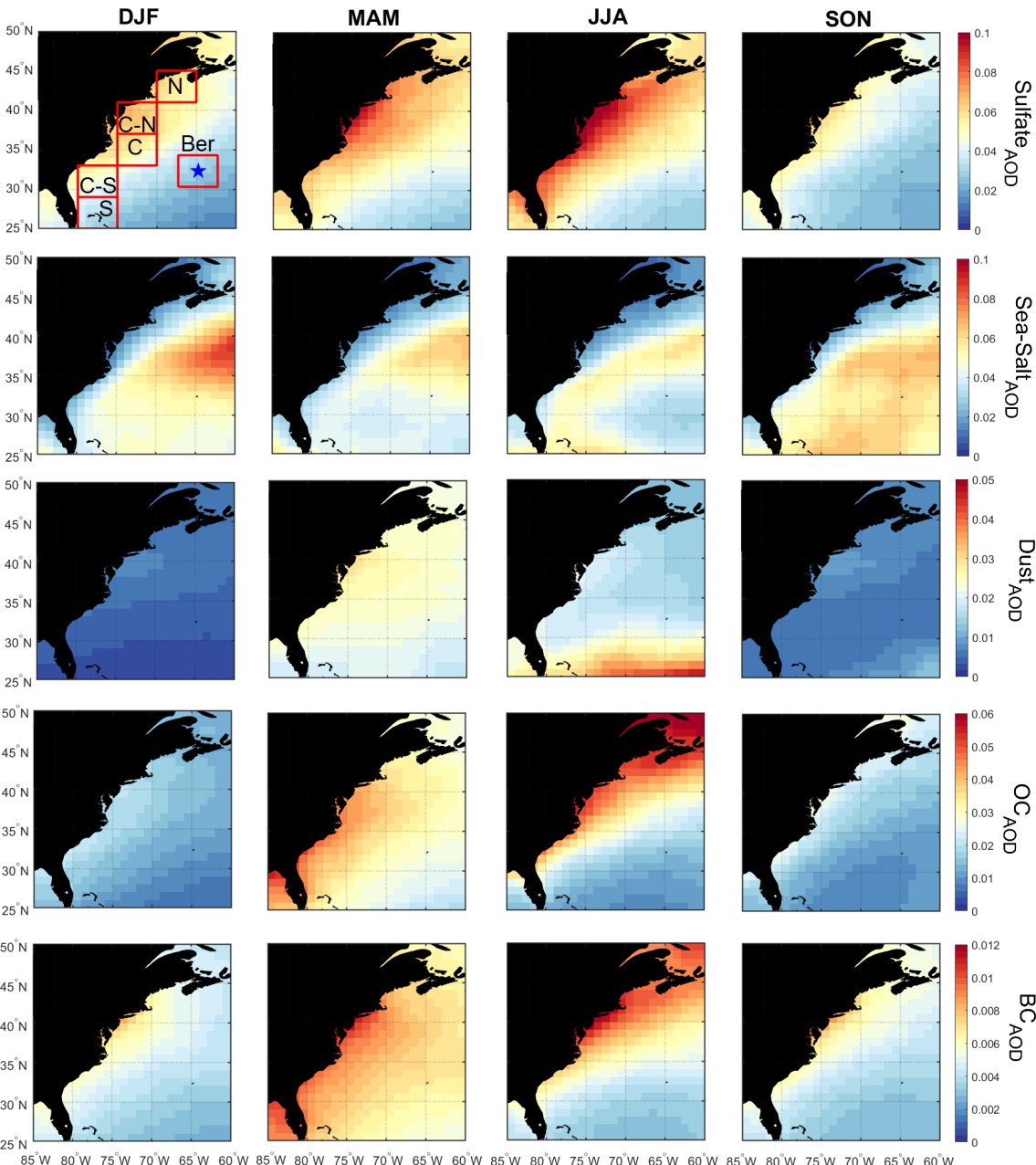

**Figure 3.** Seasonal maps of MERRA-2 speciated AOD based on data between January 2013 and December 2017. The boxes in the top left panel represent sub-domains examined in more detail throughout the study, with the blue star denoting Bermuda.

(FT). Results show that while RH is highest in JJA (except for FT of DJF in sub-domain N), differences between seasons were not very large. The maximum difference among the four seasons when considering mean RH in the PBL and FT for all sub-domains ranged between 3 %–9 % and 7 %–25 %, respectively. Consequently, humidity effects on remotely sensed aerosol parameters are less likely to be the sole explanation of the dissimilar seasonal cycle of $N_d$ and AOD, but can plausibly contribute to some extent.

One factor that could drive the seasonal variation in $N_d$ is the unwanted effects of retrieval errors in the estimation of $N_d$ at low-cloud-coverage conditions. Uncertainty associated with the estimation of $N_d$ from MODIS observation increases as cloud fraction decreases (Grosvenor et al., 2018). This is mainly because of the overestimation of droplet effective radius ($r_e$) in the retrieval algorithm due to the interference of cloud-free pixels and also high spatial inhomogeneity in low-cloud-coverage conditions that violates horizontal homogeneity assumptions in the retrieval of $r_e$ and $\tau$ from

radiative transfer modeling (Zhang et al., 2012, 2016). To test whether retrieval errors in $N_d$ are the main driver of seasonal trends, Fig. S1 shows the seasonal cycle of $N_d$ at various low-level liquid cloud fractions. The results show that as cloud fraction increases the average $N_d$ increases, regardless of season. Perhaps the more important result is that the seasonal trend in spatial maps of $N_d$ remains similar regardless of cloud fraction. This finding is important as it confirms that the seasonal cycle in $N_d$ cannot be solely explained by the uncertainties associated with the retrieval of $N_d$ at low cloud fraction.

## 3.3 Contrasting AOD and aerosol index

While previous studies have pointed to the limitations of AOD as an aerosol proxy (e.g., Stier, 2016; Gryspeerdt et al., 2017; Painemal et al., 2020), the $N_d$–AOD anticorrelation at seasonal scale over the WNAO is at odds with findings for other regions supporting the relationship between these two parameters (Nakajima et al., 2001; Sekiguchi et al., 2003; Quaas et al., 2006, 2008; Grandey and Stier, 2010; Penner et al., 2011; Gryspeerdt et al., 2016) and also that between sulfate and $N_d$ (Boucher and Lohmann, 1995; Lowenthal et al., 2004; Storelvmo et al., 2009; McCoy et al., 2017, 2018; MacDonald et al., 2020). Values of $N_d$ are influenced by the number concentration of available CCN, which is determined by aerosol properties (size distribution and composition) and supersaturation level. AOD is an imperfect CCN proxy variable because it does not provide information about composition and size distribution and is sensitive to relative humidity. Aerosol index (AI) is more closely related to CCN as it partially accounts for the size distribution of aerosols (Deuze et al., 2001; Nakajima et al., 2001; Breon et al., 2002; Hasekamp et al., 2019). The sensitivity of AI to size is evident in spatial maps for each season showing more of an offshore gradient (like sulfate AOD in Fig. 3) in each season and lacking both the offshore peak in sea salt between $\sim$ 35–40° N and the maximum AOD for dust south of 30° N in JJA. However, when comparing absolute values between the four seasons in Fig. 2b, AI exhibits a similar seasonal cycle to AOD, thereby indicating that size distribution alone cannot explain diverging seasonal cycles for $N_d$ and AOD. We next compare $N_d$ to aerosol data in the PBL where CCN more relevant to droplet activation are confined. Size distribution effects in the PBL can instead be more of a factor, especially as sea salt is abundant.

## 3.4 Aerosol size distribution and vertical aerosol distribution

Vertical profiles of aerosol extinction coefficient estimated from CALIOP nighttime observations are shown in Fig. 4 for the six sub-domains. Also shown are the seasonally representative planetary boundary layer heights (PBLHs) from MERRA-2, with numerical values of both PBLH and frac-

tional AOD contributions to the PBL and FT in Table 3. Although here we used nighttime observations from CALIOP because of having higher signal-to-noise ratio than daytime observations, we expect the general seasonal trends discussed here to remain the same regardless of the observation time. The CALIOP results indicate that aerosol extinction more closely follows the $N_d$ seasonal cycle with the highest (lowest) values in the PBL during DJF (JJA). However, aerosol extinction coefficient is sensitive to aerosol size distribution, and a plausible scenario is that DJF extinction in the PBL is primarily contributed by coarse sea salt particles, which are especially hygroscopic but do not contribute significantly to number concentration as demonstrated clearly by airborne observations (i.e., $FCDP_{>3\mu m}$ time series shown in Fig. 1d). This is supported in part by how DJF is marked by the highest fractional AOD contribution from the PBL (59 %–72 %) where sea salt is concentrated. In contrast, JJA has the lowest fractional AOD contribution from the PBL (11.3 %–52.6 %). It is also possible that the higher fractional AOD contribution from the PBL in winter is partly owed to aerosol particles being more strongly confined to the PBL compared to the summer. Sub-domains C-N and N exhibit the greatest changes in AOD fraction in the PBL between seasons with a maximum in DJF (59 %–61 %) and a minimum in JJA (11 %–19 %), suggesting they are relatively more sensitive to the aerosol vertical distribution in leading to contrasting AOD and $N_d$ seasonal cycles. Bermuda stands out as having the highest AOD fractional contributions in the PBL in DJF (72 %) and SON (69 %) and among the highest seasonal total AODs in those two seasons (0.14 in DJF and 0.10 in SON) assisted in large part by sea salt (Fig. 3) (Aldhaif et al., 2021), coincident with high seasonal wind speeds (Corral et al., 2021).

To explore aerosol number concentration characteristics in the PBL in different seasons, we next discuss results from an opportune dataset over the US East Coast (Cape Cod, MA) providing an annual profile of CCN concentration at 1 % supersaturation (Fig. 5). Cape Cod is a coastal location representative of the outflow, providing an important fraction of the CCN impacting offshore low-level clouds. As the supersaturation examined is relatively high (1 %), the measured CCN include smaller particles representing high number concentrations that would not appreciably contribute to the high aerosol extinctions from CALIOP in the PBL in direct contrast to sea salt (i.e., high extinction due to fewer but larger particles). Seasonal mean CCN values do not follow the seasonal cycle of $N_d$ nor CALIOP extinction in the PBL, with values being as follows: DJF = 1436 cm$^{-3}$; MAM = 1533 cm$^{-3}$; JJA = 1895 cm$^{-3}$; and SON = 1326 cm$^{-3}$. These results suggest the following: (i) size distribution effects are significant in the PBL when comparing extinction to number concentration, and (ii) aerosol vertical distribution behavior cannot alone explain the divergent seasonal cycles of $N_d$ and aerosol parameters (e.g., AOD, AI, surface number concentrations).

https://doi.org/10.5194/acp-21-1-2021

Atmos. Chem. Phys., 21, 1–28, 2021

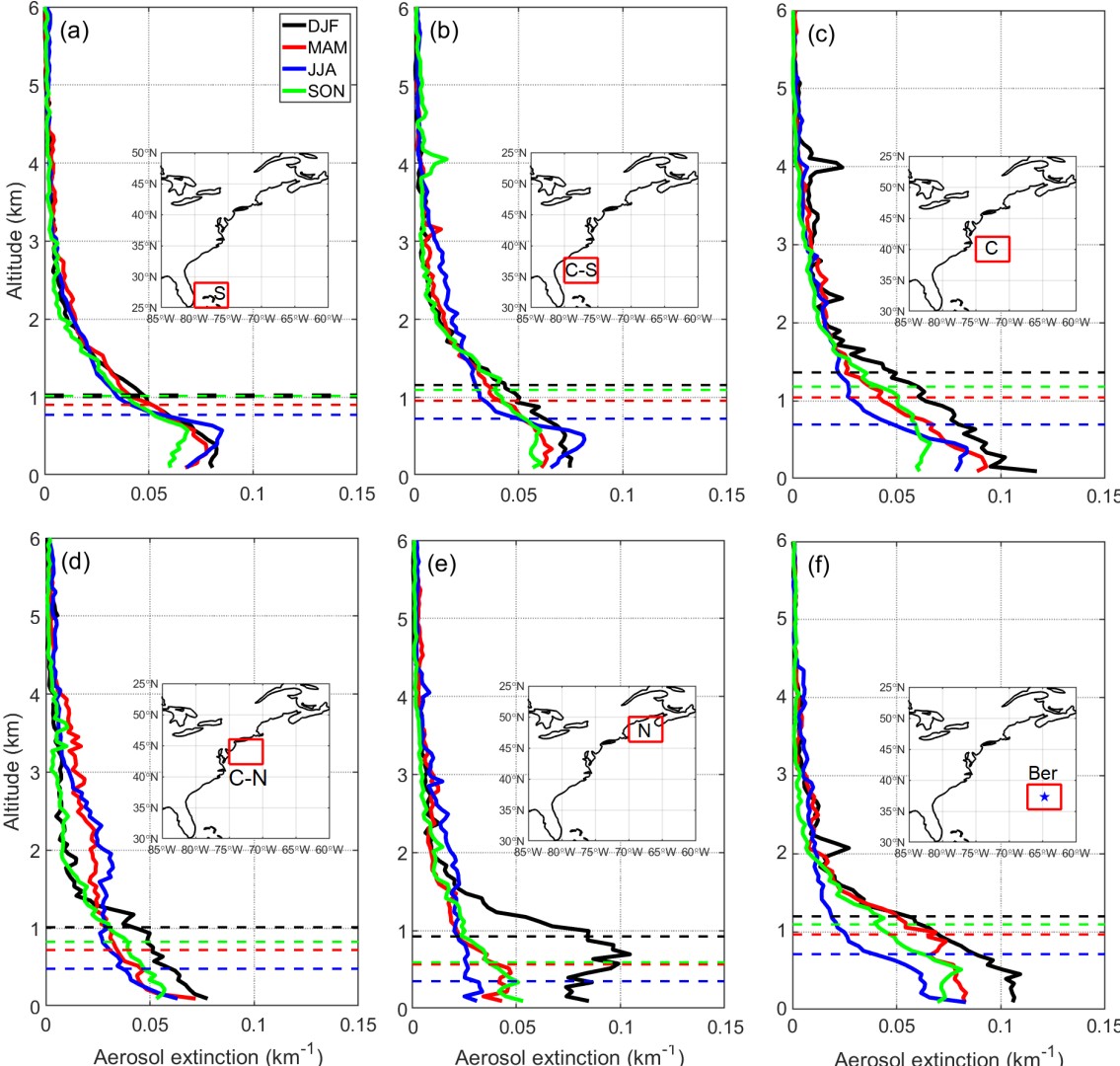

**Figure 4.** Vertical profiles of CALIPSO aerosol extinction for different seasons in **(a)**–**(f)** six different sub-domains of the WNAO. Average seasonal planetary boundary layer heights (PBLH) from MERRA-2 are denoted with dashed lines.

We next compare MERRA-2 speciated aerosol concentrations at the surface (Fig. 6) to those of speciated AOD (Fig. 3). Surface mass concentrations have the limitation of being biased by larger particles (similar to extinction). The seasonal cycle of mean values for speciated AOD and surface concentration for individual sub-domains generally agree with the exception that there was disagreement for sulfate in each sub-domain (see seasonal mean values in Table S2). Sulfate exhibited higher AODs in JJA but with surface concentrations usually being highest in DJF or MAM; although differences in seasonal mean mass concentrations were relatively small ($< 1\,\mu g\,m^{-3}$). A plausible explanation is enhanced secondary production of sulfate via oxidation of $SO_2$ or DMS convectively lifted to the free troposphere in JJA. An important result confirmed by the surface mass concentrations is that sea salt is an order of magnitude higher

than the other species, supporting the previous speculation that sea salt dominates the aerosol extinction in the PBL from CALIOP.

### 3.5 Aerosol–cloud interactions

Studies of China's east coast have shown that the aerosol indirect effect is especially strong in wintertime, whereby pollution outflow leads to high $N_d$ and suppressed precipitation (Berg et al., 2008; Bennartz et al., 2011). It is hypothesized that a similar effect is taking place off of North America's east coast, which could in part explain enhanced $N_d$ without a significant jump in aerosol parameter (e.g., AOD, AI) values necessarily. Grosvenor et al. (2018) suggested that high cloud fractions in wintertime off these east coasts relative to other seasons are coincident with strong temperature inver-

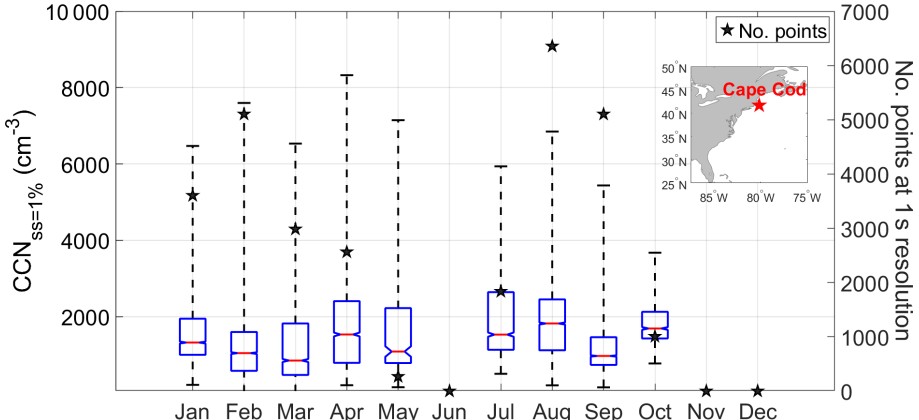

**Figure 5.** Monthly statistics of CCN concentration (1 % supersaturation) measured at Cape Cod between July 2012 and May 2013. Red lines represent the median, whiskers are the monthly range, and the top and bottom of the boxes represent the 75th and 25th percentiles, respectively. The notches in the box plots demonstrate whether medians are different from each other with 95 % confidence. Boxes with notches that do not overlap with each other have different medians with 95 % confidence.

sions usually associated with cold-air outbreaks that serve to concentrate and confine surface layer aerosols. We examine the relative seasonal strength of the aerosol indirect effect via spatial maps of the following metric commonly used in aerosol–cloud interaction (ACI) studies:

$$\mathrm{ACI} = \mathrm{d}\ln(N_{\mathrm{d}})/\mathrm{d}\ln(\alpha), \tag{3}$$

where $\alpha$ represents an aerosol proxy parameter that is represented here as AI, AOD, the speciated sulfate AOD (Sulfate$_{\mathrm{AOD}}$), and sulfate surface mass concentration (Sulfate$_{\mathrm{sf\text{-}mass}}$). The expected range by common convention is 0–1, with higher values suggestive of greater enhancement in $N_{\mathrm{d}}$ for the same increase in the aerosol proxy parameter.

Table 4 shows that DJF always exhibits the highest ACI values regardless of the aerosol proxy used, consistent with a stronger aerosol indirect effect in DJF over East Asia. The mean ACI values in DJF using AI, AOD, Sulfate$_{\mathrm{AOD}}$, and Sulfate$_{\mathrm{sf\text{-}mass}}$ ranged from 0.25 to 0.55, 0.28–0.59, 0.25–0.53, and 0.22–0.47, respectively, depending on the sub-domain. Spatial maps of ACI (Fig. 7) do not point to significant geographic features. Coefficients of determination ($R^2$) for the linear regression between $\ln(N_{\mathrm{d}})$ and $\ln(\alpha)$ when computing seasonal ACI values were generally low ($\leq 0.30$), with spatial maps of $R^2$ and data point numbers in Fig. S2. Poor correlations are suggestive of the non-linear nature of aerosol–cloud interactions (e.g., Gryspeerdt et al., 2017) and the influence of other likely factors such as dynamical processes and turbulence, data spatial resolution and dataset size, cloud adiabaticity, wet scavenging effects, and aerosol size distribution (McComiskey et al., 2009). The results of this section suggest though that aerosol indirect effects could be strongest in DJF, meaning that $N_{\mathrm{d}}$ values increase more for the same increase in aerosol. Factors that can contribute to higher ACI values in winter than summer include seasonal differences in the following: (i) dynamical processes and tur-

bulent structures of the marine boundary layer; (ii) aerosol size distributions and consequently varying particle number concentrations for a fixed mass concentration; and (iii) hygroscopicity of particles, especially as a result of changes in the composition of the carbonaceous aerosol fraction. Regarding dynamical processes and the effects of turbulence, Fig. 2 in Painemal et al. (2021) shows that heat fluxes (i.e., latent and sensible fluxes) are strongest (lowest) in the winter (summer) over the WNAO. The greater heat fluxes in DJF can contribute to more turbulent and coupled marine boundary layer conditions in winter than summer, presumably resulting in more efficient transport and activation of aerosol in the marine boundary layer, leading to higher ACI values. Forthcoming work will probe this issue in greater detail.

## 4 Discussion of potential influential factors

We probe deeper into factors related to the $N_{\mathrm{d}}$ seasonal cycle by using (Sect. 4.1) composite analyses based on high- and low-$N_{\mathrm{d}}$ days and (Sect. 4.2) advanced regression techniques tackling non-linear relationships. We focus the analyses on one sub-domain (C-N) for both simplicity and intriguing characteristics: (i) among the highest anthropogenic AOD values over the WNAO; (ii) significant seasonal changes in fractional AOD contribution to the PBL; (iii) close to the Cape Cod site where CCN data were shown; and (iv) the aerosol indirect effect (Table 4) strongest (weakest) in DJF (JJA).

### 4.1 Composite analysis

Discussion first addresses the behavior of different environmental parameters on days with the highest and lowest $N_{\mathrm{d}}$ values. Seasonal histograms of averaged daily $N_{\mathrm{d}}$ were generated for sub-domain C-N. The histograms are based on the

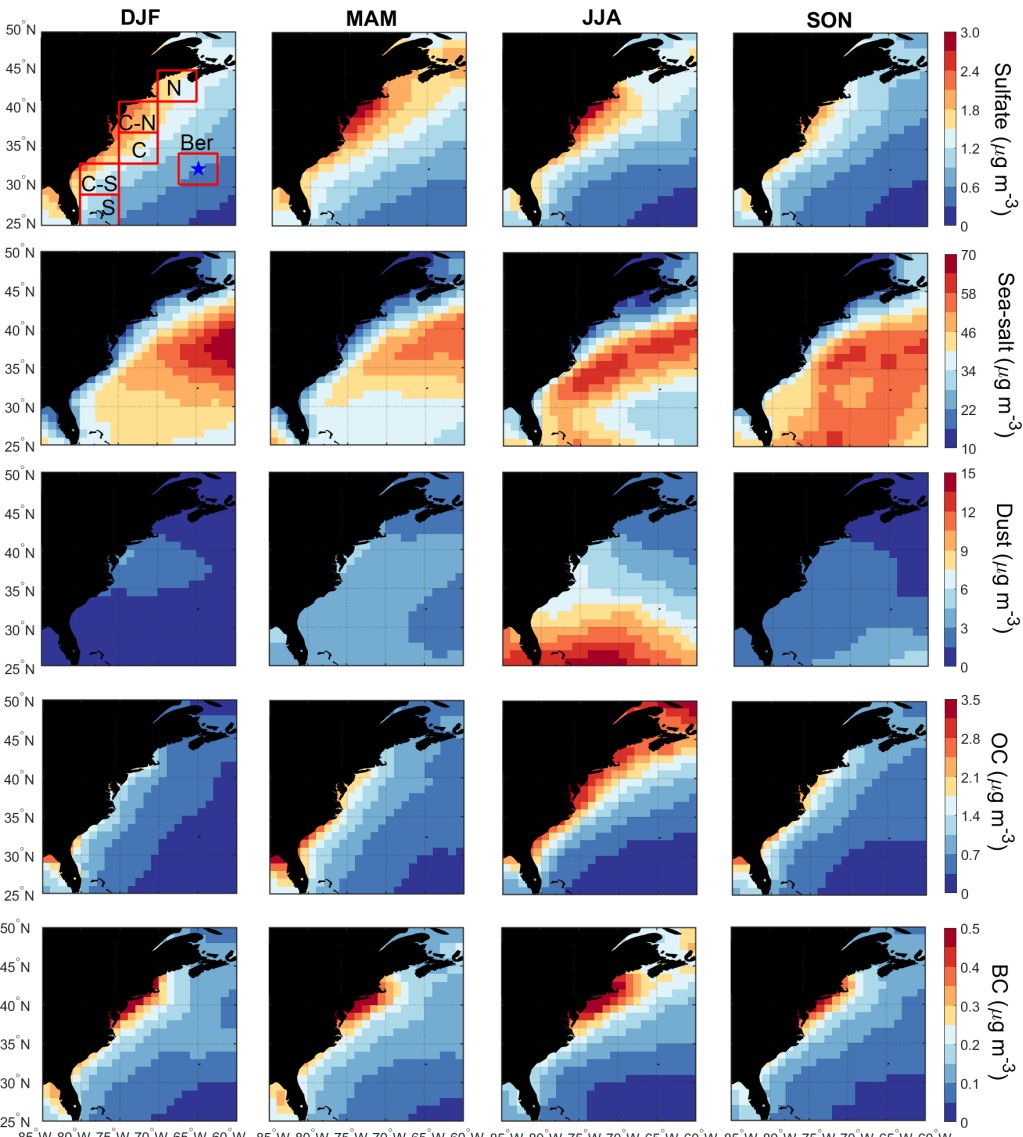

**Figure 6.** Seasonal maps of MERRA-2 speciated aerosol concentrations at the surface based on data between January 2013 and December 2017. The boxes in the top left panel represent sub-domains examined in more detail throughout the study, with the blue star denoting Bermuda.

natural logarithm of $N_d$ to better resemble a normal distribution. We assign values as being low in each season if they are less than 1 standard deviation below the seasonal value; conversely, high values are those exceeding 1 standard deviation above the seasonal mean. Cut-off $N_d$ values (cm$^{-3}$) are as follows (low/high): 33/153 (DJF), 29/118 (MAM), 38/100 (JJA), and 31/115 (SON). Next, composite maps for these groups were created (Figs. 8–12) for sea level pressure, near-surface wind, low-level cloud fraction, cold-air outbreak index, and AOD. The figures contrast the low- and high-$N_d$ maps with those showing mean seasonal values to investigate potential factors that contribute to seasonal $N_d$ variability. Interested readers are referred to Figs. S3–S20 where similar composite map results are shown for $N_d$ itself and other parameters including those in Table 2.

The resulting composite maps indicate high-$N_d$ days are characterized by (i) reduced SLP; (ii) more northerly-northwesterly flow for all seasons (except JJA) and especially stronger winds in DJF and SON; (iii) higher low-level liquid cloud fraction, especially in DJF; (iv) higher CAO index in the seasons when CAO events occur more frequently (DJF, SON, MAM); and (v) enhanced AOD. Low-$N_d$ days generally exhibited opposite conditions when compared to seasonal mean values: (i) enhanced SLP; (ii) wind ranging from southerly to westerly without any significant wind speed enhancement; (iii) reduced low-level liquid cloud fraction, es-

**Table 4.** Estimated values of ACI calculated four ways ($d\log(N_d)/d\log(AOD)$; $d\log(N_d)/d\log(AI)$; $d\log(N_d)/d\log(\text{Sulfate}_{AOD})$; $d\log(N_d)/d\log(\text{Sulfate}_{\text{sf-mass}})$) for the sub-domains shown in Fig. 2. The ACI values were obtained from log–log regression on average daily values of $N_d$ and each of the aerosol proxy variables including only the pixels with $CF_{\text{low-liq.}}$ greater than 0.1. Numbers in parentheses, in order, are $R^2$ and the number of points used for linear regression. Statistically insignificant ACI values with $p$ values greater than 0.05 are marked by bold font.

| | S | C-S | C | C-N | N | Bermuda |
|---|---|---|---|---|---|---|
| **ACI-AI** | | | | | | |
| DJF | 0.55 (0.24,440) | 0.53 (0.17,421) | 0.53 (0.14,403) | 0.33 (0.05,418) | 0.25 (0.04,403) | 0.42 (0.09,422) |
| MAM | 0.21 (0.03,451) | 0.13 (0.01,439) | 0.30 (0.06,422) | 0.17 (0.02,426) | 0.31 (0.05,428) | 0.28 (0.04,437) |
| JJA | 0.25 (0.02,437) | 0.20 (0.03,437) | 0.28 (0.07,424) | 0.11 (0.01,430) | −0.12 (0.01,408) | 0.38 (0.09,443) |
| SON | 0.23 (0.03,435) | 0.20 (0.03,428) | 0.26 (0.05,431) | 0.19 (0.04,412) | 0.24 (0.06,394) | **0.00 (0.00,428)** |
| all | 0.27 (0.05,1763) | 0.16 (0.02,1725) | 0.22 (0.04,1680) | 0.12 (0.01,1686) | 0.12 (0.01,1633) | 0.23 (0.04,1730) |
| **ACI-AOD** | | | | | | |
| DJF | 0.59 (0.13,440) | 0.53 (0.12,421) | 0.47 (0.10,403) | 0.39 (0.06,418) | 0.28 (0.04,403) | 0.37 (0.08,422) |
| MAM | 0.26 (0.02,451) | 0.22 (0.01,439) | 0.43 (0.07,422) | 0.30 (0.04,426) | 0.40 (0.06,428) | 0.32 (0.03,437) |
| JJA | **0.02 (0.00,437)** | 0.24 (0.02,437) | 0.36 (0.07,424) | 0.15 (0.01,430) | **−0.06 (0.00,408)** | 0.30 (0.04,443) |
| SON | 0.14 (0.01,435) | 0.18 (0.02,428) | 0.17 (0.02,431) | 0.16 (0.02,412) | 0.27 (0.05,394) | 0.18 (0.02,428) |
| all | 0.13 (0.01,1763) | 0.12 (0.01,1725) | 0.22 (0.03,1680) | 0.15 (0.01,1686) | 0.16 (0.02,1633) | 0.31 (0.05,1730) |
| **ACI-Sulfate$_{AOD}$** | | | | | | |
| DJF | 0.53 (0.25,440) | 0.53 (0.21,421) | 0.53 (0.19,403) | 0.37 (0.08,418) | 0.25 (0.05,403) | 0.43 (0.13,422) |
| MAM | 0.29 (0.05,451) | 0.27 (0.04,439) | 0.42 (0.14,422) | 0.32 (0.07,426) | 0.41 (0.11,428) | 0.34 (0.07,437) |
| JJA | 0.21 (0.02,437) | 0.19 (0.03,437) | 0.33 (0.09,424) | 0.20 (0.04,430) | **0.04 (0.00,408)** | 0.39 (0.09,443) |
| SON | 0.16 (0.02,435) | 0.23 (0.04,428) | 0.29 (0.07,431) | 0.28 (0.09,412) | 0.35 (0.13,394) | **0.07 (0.00,428)** |
| all | 0.23 (0.04,1763) | 0.19 (0.03,1725) | 0.30 (0.07,1680) | 0.23 (0.05,1686) | 0.22 (0.05,1633) | 0.25 (0.05,1730) |
| **ACI-Sulfate$_{\text{sf-mass}}$** | | | | | | |
| DJF | 0.44 (0.29,440) | 0.41 (0.22,421) | 0.47 (0.22,403) | 0.22 (0.04,418) | 0.23 (0.06,403) | 0.32 (0.14,422) |
| MAM | 0.24 (0.07,451) | 0.25 (0.08,439) | 0.29 (0.12,422) | 0.24 (0.05,426) | 0.36 (0.09,428) | 0.16 (0.04,437) |
| JJA | 0.11 (0.01,437) | 0.12 (0.03,437) | 0.23 (0.11,424) | 0.19 (0.06,430) | −0.12 (0.01,408) | 0.20 (0.07,443) |
| SON | 0.32 (0.16,435) | 0.36 (0.18,428) | 0.34 (0.19,431) | 0.19 (0.06,412) | 0.21 (0.05,394) | 0.17 (0.07,428) |
| all | 0.32 (0.13,1763) | 0.30 (0.12,1725) | 0.36 (0.17,1680) | 0.19 (0.04,1686) | 0.15 (0.02,1633) | 0.25 (0.11,1730) |

pecially in DJF; (iv) lower CAO index in DJF, SON, and MAM; and (v) reduced AOD in DJF and MAM, enhanced AOD in JJA, and limited change in SON. Noteworthy results from Figs. S3–S20 included the enhancement/reduction of PBLH on high-/low-$N_d$ days (least pronounced in JJA), higher/lower RH at 950 and 800 hPa on high-/low-$N_d$ days, and higher/lower sulfate AOD and surface concentrations on high-/low-$N_d$ days for DJF and MAM. Furthermore, there was a general reduction in rain on low-$N_d$ days for most seasons except SON, with rain enhancement on high-$N_d$ days except for DJF (Fig. S6); this was unexpected as wet removal was hypothesized to be a reason for reduced $N_d$ for at least the low-$N_d$ days. This may be attributed to the rain product being for surface precipitation (and thus not capturing all drizzle) and for all cloud types, including more heavily precipitating clouds deeper and higher than the low-level clouds examined for $N_d$. Another factor potentially contributing to the observed counterintuitive trends is the temporal offset be-

tween $N_d$ estimations from MODIS-Aqua and precipitation data from PERSIANN-CDR.

The mean seasonal climatological values and anomalies suggest that high-$N_d$ cases are marked by continental outflow, high cloud fractions, high PBLH, and low SLP, all of which occur most commonly in DJF and are associated with cold-air outbreaks. These events are marked by cold air over the warm ocean leading to strong surface heat fluxes, boundary layer deepening, weakened inversion strength, and high and deep clouds (Brummer, 1996; Kolstad et al., 2009; Fletcher et al., 2016; Abel et al., 2017; Naud et al., 2018). Coincident with these features is the Icelandic Low, which is a significant climatological feature of the North Atlantic whereby subpolar low pressure builds in extratropic areas beginning in the fall with westerly winds in the boundary layer that shift more to northerly in the winter (Sorooshian et al., 2020; Painemal et al., 2021). This low-pressure system seems to be stronger on high-$N_d$ days, resulting in more continental outflow and high number concentrations of CCN; the greater

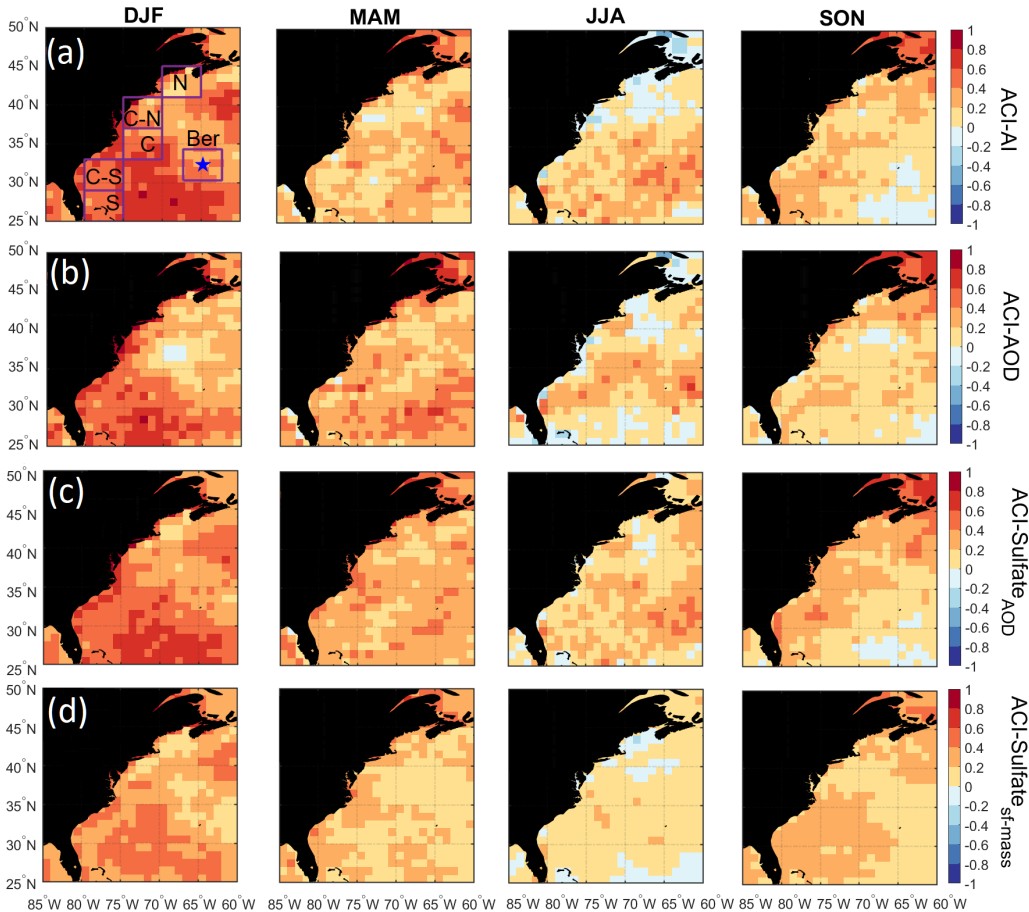

**Figure 7.** Seasonal maps of the aerosol–cloud interaction (ACI) parameters over the WNAO using daily $N_d$ and four different aerosol proxy parameters (AI, AOD, Sulfate$_{AOD}$, Sulfate$_{sf\text{-}mass}$) from CERES-MODIS and MERRA-2, respectively. ACI statistics associated with the six sub-domains shown are summarized in Table 4.

CAO index values near the coast promote high cloud coverage, affording more opportunity for cloud processing of particles to ultimately enhance droplet activation. While there can be considerable enhancement in $N_d$ as cold-air outbreak air masses evolve over warmer waters, precipitation scavenging farther downwind will be an efficient method of boundary layer aerosol (and $N_d$) removal (Abel et al., 2017; Lloyd et al., 2018), which contributes at least in part to the sharp $N_d$ gradients offshore demonstrated in Fig. 1.

### 4.2 Multivariate regression analysis

Modeling analysis focuses on the two seasons (DJF and JJA) with the extremes in terms of seasonal mean values for $N_d$ and aerosol parameters. Added motivation for examining those two seasons stems from spatial maps of $R^2$ based on ACI analysis (Fig. S2). Using the surface sulfate concentration as the aerosol proxy generally yielded higher $R^2$ values in three seasons (DJF = 0.13, MAM = 0.05, SON = 0.08) except JJA (0.02) for which the choice did not matter owing to low $R^2$ ($\leq 0.03$) values for all four aerosol proxy variables

tested. Although the $R^2$ values are all generally low, DJF and JJA are the seasons when surface sulfate levels are the most and least capable of explaining $N_d$, with $R^2$ among the four proxy variables exhibiting the widest (DJF values: 0.07–0.13) and narrowest range (JJA: 0.01–0.03) of values. We address here how much improvement is gained in modeling $N_d$ by advancing from linear regressions based on one input variable to (i) adding more input variables and (ii) moving to a more sophisticated model (GBRT) that captures non-linear relationships.

We show in Table 5 the performance of two linear models based on a single linear regression (with sulfate mass concentration) and a multi-regression that uses 14 input variables listed in Table 2. In addition, Table 5 also lists the performance of the GBRT model that ingests 14 input variables, similar to the linear multi-regression model. The average $R^2$ scores of the test set for predicting $N_d$ based on a linear regression using only sulfate surface mass concentration were 0.17 and 0.09 in DJF and JJA, respectively. In contrast, $R^2$ between the multi-regression linear model and the test dataset increased to 0.28 and 0.25 for DJF and JJA, re-

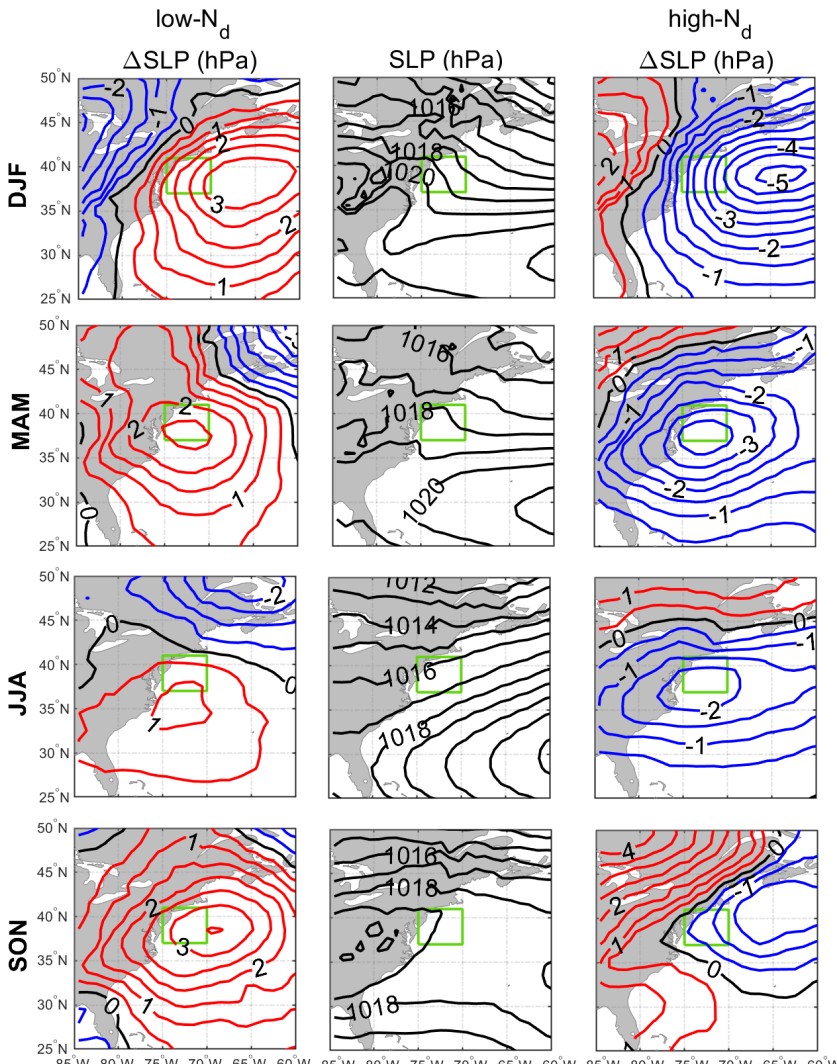

**Figure 8.** Seasonal climatology of sea level pressure (SLP) (middle column) and anomalies from seasonal averages for low-$N_d$ days (left column) and high-$N_d$ days (right column). In the left and right columns, red and blue contours are associated with positive and negative anomalies from the climatology, respectively. The green box represents sub-domain C-N for which the analysis was conducted.

**Table 5.** Performance of different models in predicting $N_d$ assessed based on average $R^2$ scores on both validation and test sets. The models were fitted separately for DJF and JJA seasons. Table 2 has the complete list of variables used in the GBRT model.

| Model | Model type | Number of predictor variables | $R^2$ score (DJF/JJA) Validation set | Test set |
|-------|------------|-------------------------------|-------------------|----------|
| $N_d \sim f(\text{Sulfate}_{\text{sf-mass}})$ | Linear | 1 | 0.17/0.09 | 0.17/0.09 |
| $N_d \sim f(\text{Sulfate}_{\text{sf-mass}}, \text{CF}_{\text{low-liq.}},....)$ | Linear | 14 | 0.27/0.24 | 0.28/0.25 |
| $N_d \sim f(\text{Sulfate}_{\text{sf-mass}}, \text{CF}_{\text{low-liq.}},....)$ | GBRT | 14 | 0.48/0.43 | 0.47/0.43 |

spectively. This increase in predictive capability was helpful to reduce the gap between seasons by presumably accounting for factors more important in JJA aside from surface concentration of sulfate. The $R^2$ scores increased even more to 0.47 and 0.43 for DJF and JJA, respectively, for the GBRT model.

Therefore, accounting for non-linear relationships improved predictive capability in both seasons. It is important to note that the GBRT model was robust in terms of overfitting and especially generalizability as $R^2$ values of the test and validation sets were similar for both seasons.

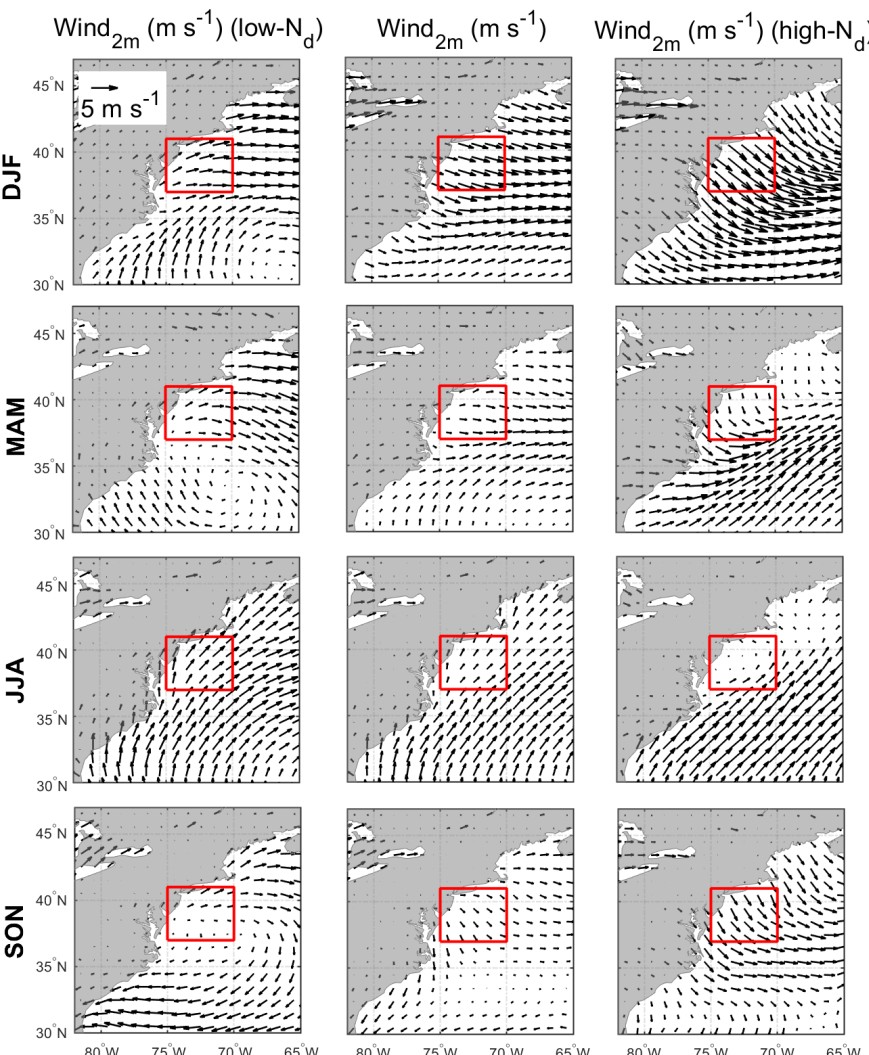

**Figure 9.** Seasonal climatology of near-surface (2 m above ground) wind speed (middle column) and mean values for low-$N_d$ days (left column) and high-$N_d$ days (right column). The reference wind vector is shown in the top left panel. The red box represents sub-domain C-N for which the analysis was conducted.

We next discuss the importance ranking of different parameters from Table 2 in terms of influencing $N_d$ for DJF and JJA (Fig. 13). Low-level liquid cloud fraction was the most important parameter in both seasons with some commonality in the next three parameters for both seasons. In DJF, sulfate surface mass concentrations were the second most important factor, followed by organic carbon surface concentrations and low-level liquid cloud-top effective height. As sulfate is secondarily produced via gas-to-particle conversion processes, this result is consistent with those from Fig. 1 showing the presumed strong impact of particles smaller than 100 nm in impacting $N_d$ values close to shore. In JJA, the CAO index was the second most important, followed by organic carbon and sulfate surface concentrations. Also, our results throughout the study and supported by modeling are in agreement with Quinn et al. (2017) that sulfate particles

contribute more to the CCN budget than sea salt particles. In DJF and JJA, the fifth most important factor was CAO index (second most important in JJA) and PBLH (11th most important in DJF), respectively.

Figures 14 and 15 show accumulated local effect (ALE) plots for the various parameters ranked in Fig. 13. In both seasons, but especially DJF, enhanced surface concentrations of sulfate and organic carbon coincide with higher $N_d$, whereas there was not any obvious positive association between $N_d$ and either sea salt or dust (Fig. 14). Dust in JJA and sea salt in DJF, seasons of which each respective aerosol type is most predominant, exhibited negative relationships with $N_d$. Such a negative relationship is plausibly related to differences between ACI when calculated using AOD versus AI (Painemal et al., 2021); for instance, coarse sea salt can expedite collision–coalescence and thus reduce $N_d$, which has

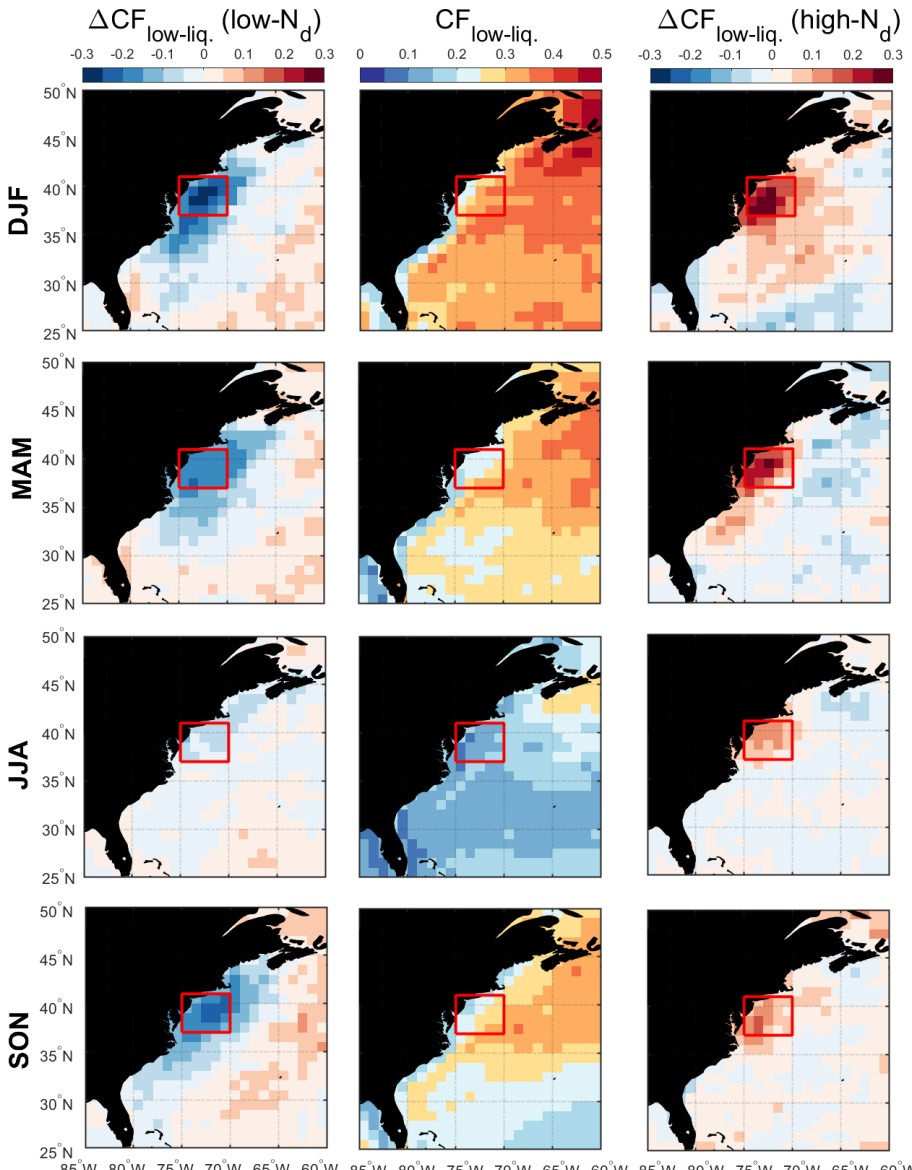

**Figure 10.** Seasonal averages of low-level liquid cloud fraction (middle column) and associated anomalies on low-$N_d$ days (left column) and high-$N_d$ days (right column). The red box represents sub-domain C-N for which the analysis was conducted.

the effect of reducing ACI (Eq. 3) and even possibly yielding negative values (Table 4). Negative values of other ACI constructs coincident with poor $R^2$ values have previously been attributed to potential effects of giant CCN (Terai et al., 2015; Dadashazar et al., 2017), but further research needs to examine this in more detail.

Figure 15 shows the similarity in the positive relationship between cloud fraction and $N_d$ in both seasons. Only in DJF did cloud-top effective height exhibit a clear relationship with $N_d$ (positive), likely linked to the common phenomenon of CAOs noted in Sect. 4.1 based on heightened CAO index values, deepening of the boundary layer, and weakened inversion strength. This is supported by enhanced $N_d$ values

coincident with negative values for $\omega_{800}$ (i.e., rising motion) and CAO index values above 0 in DJF without such relationships in JJA (Fig. 15). The six parameters in Fig. S21 (PBLH, $RH_{950}$, $RH_{800}$, Rain, $Wind_{2\,m}$, $Wind\text{-}dir_{2\,m}$) did not reveal very pronounced trends with $N_d$ in either season consistent with how they did not rank highly in importance (Fig. 13). Of particular interest is $Wind_{2\,m}$, which is used here as a proxy variable for updraft speed in the marine boundary layer, which is expected to have a high impact on $N_d$ via its effect on in-cloud supersaturation. Although the ALE plot of $Wind_{2\,m}$ suggested a small increase of about $\sim 10\,\text{cm}^{-3}$ in $N_d$ as the wind speed increased, $Wind_{2\,m}$ did not come out as a very important parameter in either season. This may

https://doi.org/10.5194/acp-21-1-2021

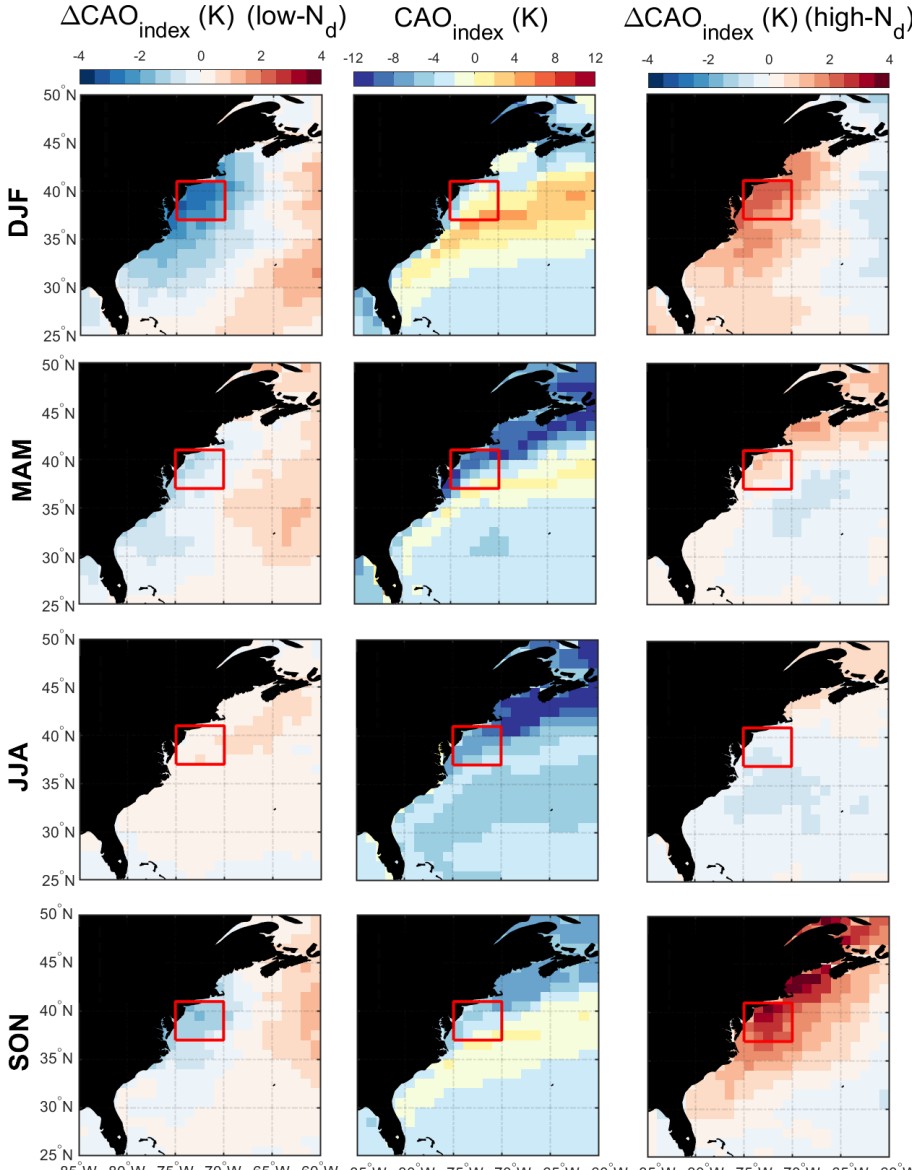

**Figure 11.** Seasonal averages of cold-air outbreak (CAO) index (middle column) and associated anomalies on low-$N_d$ days (left column) and high-$N_d$ days (right column). The red box represents sub-domain C-N for which the analysis was conducted.

be due to the fact that environmental conditions representing updraft speed were already included in parameters such as cloud fraction and CAO index. Another explanation can be the shortcomings and high uncertainties associated with the use of Wind$_{2\,m}$ as a proxy for updraft speed.

The results of regression analysis highlight the high sensitivity of $N_d$ to cloud fraction regardless of season. As discussed earlier, this can be attributed largely to two factors: (i) the relationship between cloud type (e.g., stratocumulus, shallow cumulus) and cloud fraction, which can, in turn, influence cloud microphysical properties like $N_d$, and (ii) uncertainties associated with $N_d$ estimates from satellite observations that can result in negative biases in $N_d$ for low-cloud-

coverage conditions. To further test the relative influence of various variables at different cloud fractions, two sensitivity tests with GBRT modeling were conducted using subsets of data with varying cloud fraction ($0.2 \leq CF_{\text{low-liq.}} \leq 0.4$ and $CF_{\text{low-liq.}} \geq 0.7$).

Beginning with results for $CF_{\text{low-liq.}} \geq 0.7$ (Figs. S22–S25), the average $R^2$ scores for validation and test sets for these runs were 0.47/0.39 (DJF/JJA) and 0.49/0.38 (DJF/JJA), respectively. A feature that stands out is that for both DJF and JJA, surface mass concentrations of sulfate became the most important factor. ALE plots presented in Fig. S23 suggest that $N_d$ has a very similar sensitivity to sulfate concentration in high-cloud-coverage conditions regard-

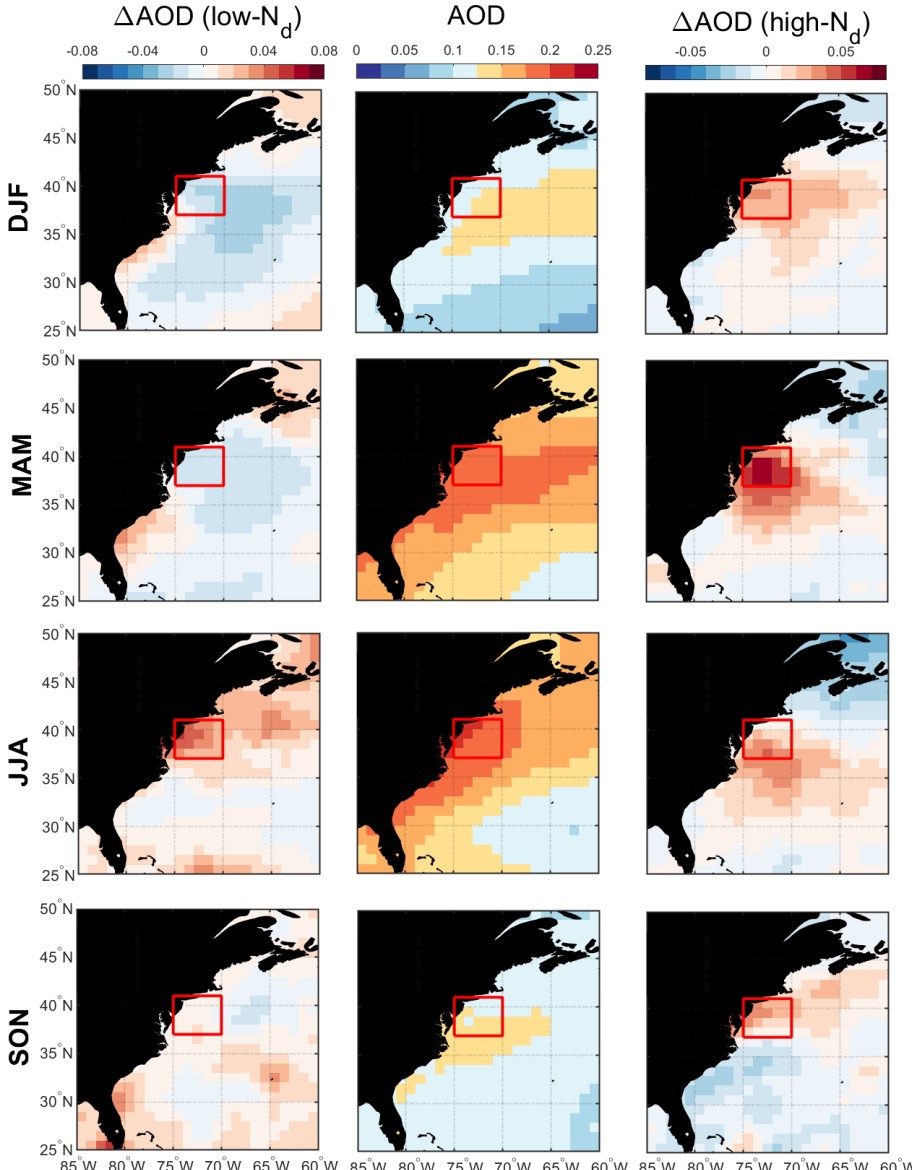

**Figure 12.** Seasonal averages of MERRA-2 AOD (middle column) and associated anomalies on low-$N_d$ days (left column) and high-$N_d$ days (right column). The red box represents sub-domain C-N for which the analysis was conducted.

less of season in contrast to the results of the original run where $N_d$ was more sensitive to the changes in sulfate level in DJF than JJA. These results are in agreement with previous studies where $N_d$ values for marine boundary layer clouds were highly sensitive to sulfate concentrations at the level close to cloud base (Boucher and Lohmann, 1995; Lowenthal et al., 2004; Storelvmo et al., 2009; McCoy et al., 2017, 2018; MacDonald et al., 2020). The second most important factor for DJF was the surface mass concentrations of organic carbon followed by $CF_{low-liq.}$ and sea salt surface mass concentrations. On the other hand, the second most important factor in JJA was CAO index followed by $CF_{low-liq.}$ and wind direction. ALE plots presented in Figs. S23–S25 showed similar

relationships between $N_d$ and input parameters as observed for the original runs where full datasets were used as the input.

Figure S26 shows the results of the GBRT model using input data with a cloud fraction between 0.2 and 0.4, the condition relatively more prevalent in JJA than DJF. The average $R^2$ scores for validation and test sets for these runs were 0.30/0.30 (DJF/JJA) and 0.33/0.31 (DJF/JJA), respectively. It is interesting to see that for both seasons, an aerosol parameter emerged as the most important factor. Mass concentrations of OC appeared as the most important factor in JJA (the fourth most important factor in DJF) while in DJF, sulfate concentration came out as the most important factor

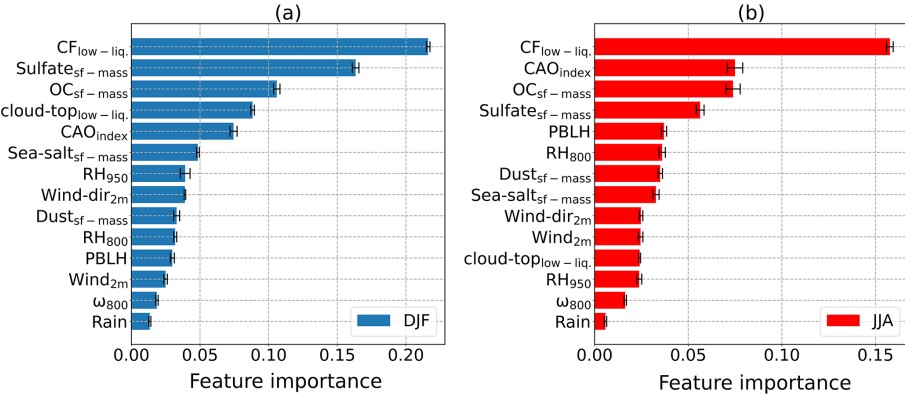

**Figure 13.** Average permutation feature importance of input parameters for **(a)** DJF and **(b)** JJA based on GBRT models trained in each season. Feature importance values were calculated based on using the test set. Error bars exhibit the range of feature importance values stemming from the variability of the obtained models from the cross-validation resampling procedure.

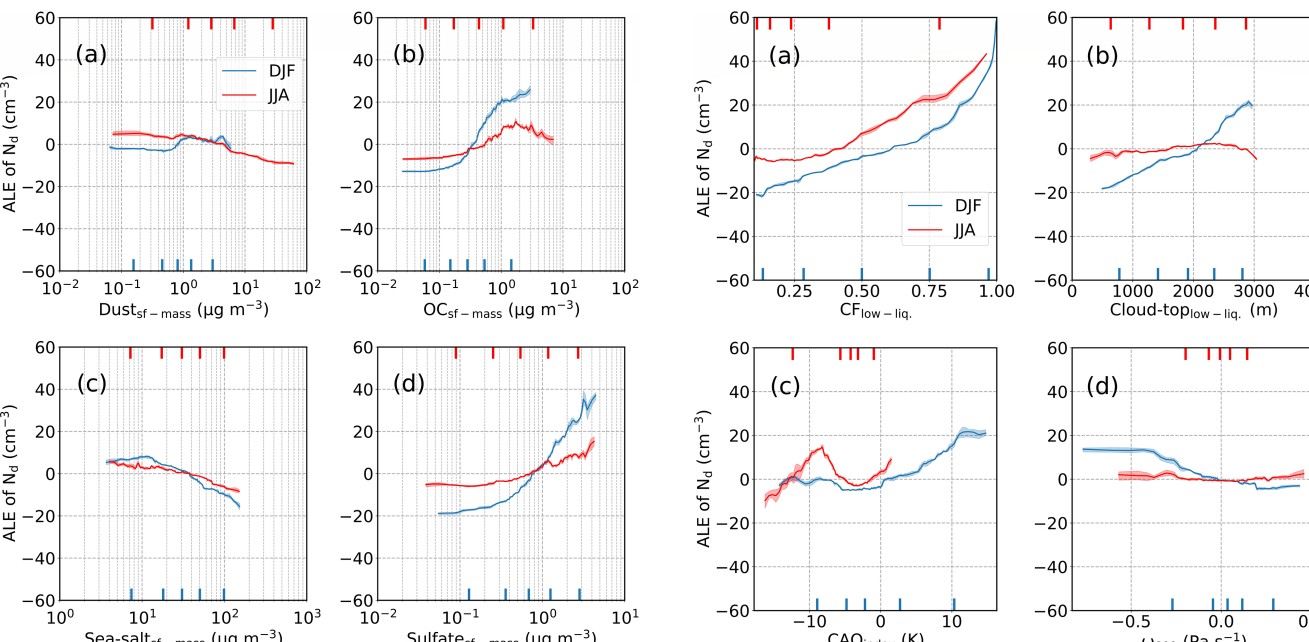

**Figure 14.** Average local accumulated effect (ALE) profiles based on GBRT modeling for surface mass concentrations of the following parameters: **(a)** dust, **(b)** organic carbon, **(c)** sea salt, and **(d)** sulfate. Blue and red profiles represent ALEs of DJF and JJA, respectively. Shaded areas show the ALE ranges stemming from the variability of the obtained models from the cross-validation resampling procedure. Markers on the bottom and top $x$ axes denote the values of the 5th, 25th, 50th, 75th, and 95th percentiles for each input variable.

(the fourth most important factor in JJA) consistent with the results of previously discussed models for DJF. It should be noted that ALE plots also suggested less sensitivity of $N_d$ to sulfate in JJA than DJF, similar to the results observed in the original model run including all the data points. The second most important factor in DJF turned out to be the cloud-

**Figure 15.** Same as Fig. 14 but for the following input parameters: **(a)** low-level liquid cloud fraction (CF$_{\text{low-liq.}}$), **(b)** cloud-top effective height of low-level liquid cloud (cloud-top$_{\text{low-liq.}}$), **(c)** cold-air outbreak (CAO) index, and **(d)** vertical pressure velocity at 800 hPa ($\omega_{800}$).

top effective height of low-level liquid clouds followed by CAO index. On the other hand, CAO index was the second most important factor in JJA followed by PBLH. ALE plots presented in Figs. S27–S29 also showed similar qualitative trends observed in original and high-cloud-coverage runs.

## 4.3 Unexplored factors

Additional factors impacting the relationship between aerosol and $N_d$ seasonal cycles are discussed here that war-

rant additional research with more detailed data at finer scales such as with aircraft. We are cognizant that this list is not fully exhaustive. As low-level cloud fraction impacted model results of Sect. 4.2 so substantially, the dynamics of the studied clouds require further characterization. As cloud fraction and CAO index are well related, especially in DJF, aerosol–cloud interactions are likely stronger than other seasons (as implied by Sect. 3.5) due in part to enhanced surface fluxes and turbulence and thus more droplet activation with higher cloud supersaturations (Painemal et al., 2021); in contrast, the smaller shallow cumulus clouds in summertime may be less favorable for droplet activation due to factors such as reduced turbulence and more lateral entrainment.

Entrainment of free tropospheric aerosol can impact $N_d$ values, with potentially varying degrees of influence between seasons. It is presumed that with summertime convection, the more broken cumulus scenes are less adiabatic through the cloudy column and more affected by entrainment and mixing; hence, $N_d$ values derived using data that remote sensors retrieve near cloud top could be considerably lower than values lower by cloud base. Satellite remote sensing studies of aerosol–cloud interactions will presumably be more challenging in winter periods versus the summer with regard to the spatial and temporal mismatch between cloud and aerosol retrievals. More specifically, it is easier to get nearly coincidental sampling in summertime due to lower cloud fractions, while in winter the frontal regions with high cloud fractions make it challenging to get aerosol retrievals. There is complexity in understanding how aerosols relate to $N_d$ due to how giant CCN can reduce $N_d$ and also since wet scavenging can remove aerosols efficiently. As aircraft data are limited and difficult to use for assessing seasonal cycles, new techniques of retrieving CCN and $N_d$ from space will greatly assist such types of studies in the future.

## 5 Conclusions

This work investigates the seasonal cycle of $N_d$ over the WNAO region in terms of concentration statistics and with discussion of potential influential factors. The results of this work have implications for increased understanding of aerosol–cloud interactions and meteorological factors influencing concentration of cloud droplets in the marine boundary layer. The results and interpretations can be summarized as follows in the order of how they were presented.

- An ACTIVATE case flight during the DJF season shows a sharp offshore $N_d$ gradient ranging from $> 1000$ to $< 50\,cm^{-3}$ explained in part by particles smaller than $100\,nm$ activating into drops during a cold-air outbreak with post-frontal clouds. There were significant changes in aerosol composition in cloud-free air and also in droplet residual particles as a function of offshore distance. These changes included a sharp decrease in aerosol number concentration, a decrease in mass

fraction of sulfate in droplet residual particles, and an increase in mass fraction of organic and chloride of droplet residual particles moving offshore.

- $N_d$ is generally highest (lowest) in DJF (JJA) over the WNAO, but aerosol parameters such as AOD, AI, surface-based aerosol mass concentrations for most species, and CCN concentrations (1 % supersaturation) are generally highest in JJA and MAM and are at (or near) their lowest values in DJF. While aerosol extinction in the PBL is highest in DJF, it is driven largely by sea salt (large but few in number) and thus cannot explain the $N_d$ peak in wintertime.

- While relative humidity was generally highest in JJA across the WNAO, the differences between seasons in the PBL and FT were not sufficiently large to explain the divergent seasonal cycles of AOD and $N_d$.

- The susceptibility of $N_d$ to aerosols (Eq. 3) was strongest in DJF using four different proxy variables for aerosols, suggestive of at least one reason why $N_d$ can be highest when aerosol proxy variables for concentration are typically near or at their lowest values.

- Composite maps of high- versus low-$N_d$ days across the WNAO reveal that conditions associated with the highest-$N_d$ days, regardless of season (but especially DJF), are reduced sea level pressure, stronger winds aligned with continental outflow, high low-level liquid cloud fraction, higher CAO index and PBLH, and enhanced AOD. Cold-air outbreaks are coincident with all of these conditions, especially in the colder months of DJF in sharp contrast to JJA when $N_d$ is lowest.

- Gradient-boosted regression analysis shows that the most important predictors of $N_d$ in DJF and JJA vary to some extent, but with cloud fraction being the most important parameter, followed by either (for DJF) surface mass concentrations of sulfate and organic carbon and CAO index or (for JJA) CAO index, surface mass concentrations of organic carbon, and sulfate concentrations. Accumulated local effect plots confirm that sulfate and organics help drive the high $N_d$ values via continental outflow, which is assisted in large part by conditions associated with CAOs such as high cloud fraction and high CAO index.

Therefore, the combination of continental pollution outflow and turbulence changes contributed by surface fluxes (manifested in the strongest CAO index values in DJF and weakest in JJA) markedly influence the $N_d$ cycle, leading to differing annual cycles in cloud microphysics and aerosols. More detailed data such as from aircraft and modeling can help extend this line of research to confirm these findings and speculations such as how (i) the aerosol indirect effect is strongest in DJF due to boundary layer dynamics such as

with more turbulence and mixing than other seasons (Painemal et al., 2021); (ii) enhanced giant CCN in forms such as sea salt and dust can reduce $N_d$ via expediting the collision–coalescence process; and (iii) substantial aerosol removal can occur far offshore as postfrontal clouds associated with CAOs build and then begin to precipitate. The latter hypothesis may help explain why Bermuda ($> 1000$ km offshore the US East Coast) was the only selected sub-domain in this study to not have a seasonal $N_d$ peak in DJF.

*Code availability.* ALE plots were calculated using the source code provided by Jumelle et al. (2021) available at https://github.com/blent-ai/ALEPython.

*Data availability.* All the data used in this study are open access and can be found online as follows. CERES-MODIS data as described by Minnis et al. (2011) are available at https://ceres.larc.nasa.gov/data/; CALIPSO data are available at https://subset.larc.nasa.gov/calipso/ (CALIPSO, 2021); PERSIANN-CDR data as described by Nguyen et al. (2019) are available at https://chrsdata.eng.uci.edu/; MERRA-2 data are available at https://disc.gsfc.nasa.gov/ (GMAO, 2021); TCAP CCN data as described by Berg et al. (2016) are available at https://adc.arm.gov/discovery; ACTIVATE Airborne Data are available at https://doi.org/10.5067/ASDC/ACTIVATE_Aerosol_AircraftInSitu_Falcon_Data_1 (NASA/LARC/SD/ASDC, 2020a), https://doi.org/10.5067/ASDC/ACTIVATE_Cloud_AircraftInSitu_Falcon_Data_1 (NASA/LARC/SD/ASDC, 2020b), and https://doi.org/10.5067/ASDC/ACTIVATE_MetNav_AircraftInSitu_Falcon_Data_1 (NASA/LARC/SD/ASDC, 2020c).

*Supplement.* The supplement related to this article is available online at: https://doi.org/10.5194/acp-21-1-2021-supplement. TS5

*Author contributions.* HD, DP, and MA conducted the analysis. AS and HD prepared the manuscript. DP assisted with CERES-MODIS data usage. MA performed the CALIPSO analysis, and HD carried out all other data analysis. EC, SK, RHM TS6, CR, MS, KT, CV, EW, and LZ participated in airborne data collection. AS and HD prepared the manuscript with other authors contributing with editing support. TS7

*Competing interests.* The authors declare that they have no conflict of interest.

*Special issue statement.* This article is part of the special issue "Marine aerosols, trace gases, and clouds over the North Atlantic (ACP/AMT inter-journal SI)". It is not associated with a conference.

*Acknowledgements.* The work was funded by NASA grant 80NSSC19K0442 in support of ACTIVATE, a NASA Earth Venture Suborbital-3 (EVS-3) investigation funded by NASA's Earth Science Division and managed through the Earth System Science Pathfinder Program Office. The authors acknowledge the NOAA Air Resources Laboratory (ARL) for the provision of the HYSPLIT transport and dispersion model and READY website (http://ready.arl.noaa.gov, last access: 14 June 2021) used in this work.

*Financial support.* This research has been supported by the National Aeronautics and Space Administration (grant no. 80NSSC19K0442).

*Review statement.* This paper was edited by Stefania Gilardoni and reviewed by two anonymous referees.

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

## Remarks from the typesetter