# Peer review of "Cloud Drop Number Concentrations over the Western North Atlantic Ocean: Seasonal 2 Cycle, Aerosol Interrelationships, and Other Influential Factors 3 Hossein Dadashazar1, David Painemal2,3, Majid Alipanah4, Michael Brunke5, Seethala 4 Chellappan6</s"

_Atmospheric Chemistry and Physics, 2021_

## Referee Comment (RC1)

**Review of the manuscript:**
**"Cloud Drop Number Concentrations over the Western North Atlantic Ocean: Seasonal Cycle, Aerosol Interrelationships, and Other Influential Factors"**
**by Dadashazar et al.**

**General Comment**

In the paper "Cloud Drop Number Concentrations over the Western North Atlantic Ocean: Seasonal Cycle, Aerosol Interrelationships, and Other Influential Factors" by Dadashazar et al., the authors investigate cloud droplet number concentrations (Nd) and its influential factors on multiple scales on the basis of diverse observational data sets. The analysis first describes high-resolution aircraft measurements of a flight from the ACTIVATE campaign, and puts these high-resolution measurements into a wider context with the description of the general seasonal cycles of cloud/aerosol/meteorology of the region. The authors go into the details of aerosol size and vertical distributions and calculate aerosol-cloud-interaction (ACI) statistics in different seasons. Potential influential meteorological factors are described in two analyses, using a composite approach by contrasting high and low Nd days, and by the application of a machine learning algorithm. Some of the main findings of the paper are that ACI is generally strongest during DJF, when Nd values tend to be highest, and that high Nd days are shown to feature systematically different meteorology (e.g. stronger continental outflow) when compared to low Nd days.

The topic of the paper is highly relevant to the aerosol/cloud/climate community and of high interest to the readership of ACP. The paper presents a thorough analysis of comprehensive observational data sets that advances the scientific understanding of the observed Nd patterns of the region. The paper is well written and structured and displays high-quality figures. I have only some minor points the authors need to address and some specific remarks that the authors may want to consider. I therefore recommend the manuscript for publication in ACP after minor revisions.

**1   Minor Points**

1. I think the authors should discuss vertical velocity as one of the main drivers of Nd variability in some more detail. The authors do this to some extent already in the manuscript, but I think it is necessary to point out that this is likely an important factor which is e.g. not provided as an input to the GBRT models. I would suggest that the authors include near-surface wind speed as a proxy for boundary layer

turbulence/updrafts in the GBRT, especially since winds seem to be an important factor in the composite analysis.

2. In section 2.3 I am missing information on the hyperparameters of the GBRT models and how these are tuned during the training/validation phase. This is critical to be able to reproduce the results and informative for readers interested in the technical details of the model setup. In my opinion, this information could be provided for in a table and may be best suited in the supplement, though.

3. What is the temporal relationship between the Nd and precipitation data? I believe it would be good to a) describe the time of satellite observations in subsection 2.2.1 and the precipitation data in 2.2.3 and b) discuss the potential influence of temporal offsets for the purpose of analyzing wet scavenging effects. I am also wondering why the authors did not chose to use information on precipitation from Cloudsat given they already use A-train data.

4. I think it would beneficial to briefly comment if the vertical distribution of aerosols at nighttime (used here) is excepted to be significantly different from the daytime (rest of the data sets analyzed here).

5. This is just to initiate discussion: In the air-quality community, it is well known that the seasonal cycle of satellite-AOD and near-surface particle concentrations over continental regions frequently show contrasting seasonal cycles (e.g. Koelemeijer et al. 2006, 10.1016/j.atmosenv.2006.04.044). This is caused by effects of PBLH and BL humidity (Stirnberg et al. 2018, 10.3390/rs10091353), and can be corrected for to some degree (Arvani et al. 2016, 10.1016/j.atmosenv.2016.06.037). I believe that at least qualitatively there is something to learn from these findings that have implications for the ACI community (and this paper specifically) as well, especially in studies covering continental regions or regions of strong continental outflow. I don't think the authors have to discuss this in their paper, but it may be a useful discussion to have in the ACI community and in my opinion links well to the findings presented here. The authors may chose ignore, comment or discuss this point as they find most useful.

**2 Specific Remarks**

l. 58 In my opinion, the word "potentially" does not apply to "enhanced cloud albedo" in the case of increased Nd and constant LWP, but only applies to the latter two points.

l. 206 There is a typo: "supermicromemter"

l. 209 Please use SI units (760 torrs)

l. 385 I think Stier (2016, 10.5194/acp-16-6595-2016) is a relevant source that should be cited here.

l. 482 "usually always" - please remove one of them

---

## Referee Comment (RC2)

This paper attempts to explain the seasonality of cloud drop number concentrations (Nd) off the coast of the USA and Canada with a focus on explaining why Nd is highest in the DJF season, but the aerosol optical depth (AOD) is lowest in that season. The paper presents some useful analysis and I think it should be published once the concerns have been addressed. However, there are places where the results are not fully explained, some places where there should be a more quantitative analysis and some results that are in the supplementary that should be in the main text. Some of the arguments could also be made more clearly. It seems that aerosol is more efficient at making cloud droplets in DJF, seemingly because of the prevalence of the trade cumulus and stratocumulus conditions (with high low-altitude cloud fractions). However, the paper doesn't quite get the bottom of why this should be – perhaps more discussion on this is warranted although it may be case that we don't quite know yet. Here are some suggestions that might help to get closer to an answer :-

- It would be interesting to examine what the important predictors are in the subset of data with high low altitude cloud fractions. E.g., do the aerosol parameters then become equally important between the seasons once we are in the cumulus/stratocumulus regime (particularly CCN and sulphate surface mass)? Is Nd similar in DJF and JJA for the high cloud fraction subset indicating that it is mainly the prevalence of the low cloud conditions in DJF that cause the Nd difference? You could also do a similar analysis for the subset with small low altitude cloud fractions (more typical of JJA perhaps).

- Could it be the case that the Nd retrievals don't work very well and give smaller Nd values when there are no boundary layer clouds (since they are designed to look at PBL liquid clouds)? This could then give higher Nd in the conditions with more PBL clouds (i.e., DJF). There is also likely to be more overlying higher altitude cloud in JJA, which would affect the retrieval of Nd. It would be good to look at the types of clouds and situations in which Nd is being retrieved in JJA – i.e., whether most of the retrievals come from mid-level clouds, or clouds with overlying ice cloud, etc.

- Can the composite analysis be done with the Cape Cod CCN data? Or can the CCN be used in the ML model? E.g., is it a better predictor than surface sulphate mass?

- Can you look at boundary layer decoupling index? Perhaps the high cloud fraction regime in winter is more coupled than the summer regime allowing more efficient transport of the surface aerosol to the clouds?

- It seems that the offshore flow prevalent in DJF (cf. southeasterlies from the open ocean in JJA) might also play some role in determining Nd since it may transport more CCN from the continent? Are there more measurements that can help to elucidate whether this may be the case (e.g., aerosol size distribution data or additional CCN data from the Cape Cod data)? It might be good to include the wind direction as a predictor in the ML?

- Other possible reasons for the DJF aerosols (or aerosols when there is lots of PBL cloud) being more efficient at making droplets could be discussed. E.g., could the aerosol be more hygroscopic in these conditions, could there be higher number concentrations (since here you mainly consider mass concentrations). These difference may be related to the different aerosol sources due to the different wind direction (see previous point).

- Why did you not include AOD, speciated AOD, speciated boundary layer AOD, etc., in the ML model so that their impact relative to sulphate surface mass, etc. can be quantified?

Also some more line specific comments :-

Section 3.1 – I'm not sure how well this section works where it is, or how useful the aircraft analysis is for the main conclusions of the paper. Maybe it would be better placed at the end of all the other results in order to help highlight some of the issues raised in the rest of the paper?

L318 – Is it possible to calculate an approximate activation diameter for 0.43% supersaturation given the other aircraft measurements? This would make it easier to compare to the Dp>10 and Dp>3um data.

L381 – "Consequently, humidity effects on remotely sensed aerosol parameters cannot alone explain the dissimilar seasonal cycle of Nd and AOD, but can plausibly contribute to some extent."

- You haven't proved this quantitatively. Can you do a calculation of how much impact the RH difference would have?

L411 – "but do not contribute significantly to number concentration as demonstrated clearly by airborne observations (Figure 1)."

- It's not clear which part of Fig. 1 demonstrates this? Can you explicitly point this out?

L413 – "This is supported in part by how DJF is marked by the highest fractional AOD contribution from the PBL (59 – 72%) where sea salt is concentrated. In contrast, JJA has the lowest fractional AOD contribution from the PBL (11.3 – 52.6%)."

- But this could also indicate higher CCN concentrations in the PBL in DJF perhaps due to the aerosol being more trapped there than in summer?

-

Figure 3 – It looks like here the PBL AOD would be higher in DJF than for JJA for many of the regions. It would be good to quote the PBL AOD values in Table 3. You need to describe these results in more detail - the enhanced PBL values in DJF are an important point to describe even though it seems that it doesn't explain the Nd seasonality.

L447 – "We next compare MERRA-2 speciated aerosol concentrations at the surface (Figure S2) to those of speciated AOD (Figure S1)."

- I think that this information is very important and should not be in the supplementary since whether the PBL aerosol mass concentrations (or ideally CCN number concentrations, but I think they are not available from MERRA?) are lower in DJF compared to JJA is a key part of the analysis regarding why the DJF AOD is lower and yet the Nd higher compared to JJA. Indeed, Fig. S2 suggests that surface sulfate mass concentration is about the same in DJF and JJA despite the AOD difference (and sulfate AOD is also higher in JJA than DJF according to MERRA). Of course we might expect sulfate mass concentrations to be even higher in DJF than in JJA if it was to explain the Nd difference, but it does suggest that part of the issue is that AOD is vertically integrated and is not just for the PBL.

- Figs. S1 and S2 also suggest that near the coast sulfate AOD dominates over sea-salt, which argues against the higher observed (Fig. 3) PBL extinction values in DJF being due to sea-salt (as argued e.g., L413 and L707).
- Although sea-salt surface mass concentrations are higher than sulfate. Do you have speciated profiles of extinction from MERRA? These could be used to quantify the effect of sea-salt on the PBL AOD.
- What seems a bit strange given that DJF and JJA have similar surface SO4 in MERRA is that the Cape Cod observations show lower CCN concentrations in DJF. It would also be good to discuss this a bit more along with potential caveats. You mention that the 1% supersaturation at which the CCN are measured is quite high and would be counting fairly small aerosol particles – I think it would be good to show the lower supersaturations that you say are available. Or at least check whether the DJF values of these are also lower than in JJA (data permitting). Is there any other CCN data down the east coast of US since it would be very useful here. Or do you have observed aerosol size and composition measurements that might help determine whether the supersaturation has a big effect and whether there really are fewer CCN in DJF?
  - Also, how representative is the 1-year of data likely to be? Could the interannual variability be large enough to make that result uncertain?
  - It could also be that MERRA is doing a poor job of representing the sulphate mass concentration.
  - Finally, it would be good to quantify how likely it is that we can have a similar surface sulphate mass concentration, but different CCN (using observations or a more sophisticated model perhaps).
- So overall I think that the analysis discussion on the above issues can be improved quite a bit.
-

**Typos / grammar etc.**

L252 – "The ALE value of feature S" – it's not clear here what you mean by "feature". The symbols for this equation have also not all been explained. What does subscript c refer to? What are $x_s$, $x_c$, $z_s$? "The value of $f_{s,ALE}(x_s)$ can be viewed as the difference between the model's response at $x_s$ and the average prediction." – this is not very clear – I think this explanation needs to be clearer for interpreting the later figures.

L288 – the flight leg abbreviation "BCT1" is stated here without definition. It would be good to have introduced the different flight legs being described before this with reference to Fig. 1.

L305 – "ranged" -> "ranged from".

L307 – "which is a fairer comparison with the ACB1 leg" – fairer than what?

L311 – "there was a significant offshore gradient in LAS submicrometer particle number concentration and AMS non-refractory aerosol mass, ranging from 424 cm-3 and 5.60 µg m-3 (from

BCB1) to 21 cm-3 and 0.32 μg m-3 (from BCB3), respectively; these values are based on times of the maximum and minimum LAS concentrations during the BCB1 and BCB3 legs, respectively."

- This sentence was a little confusing and could be made clearer I think.

L320 – "There was a slighter gradient in particle concentrations with Dp > 3 μm (most likely sea salt) between the same two points of maximum and minimum LAS concentration in BCB1 and BCB3 legs, respectively: 0.26 cm-3 to 0.11 cm-3."

- This could be written more clearly.

Fig. 4 – "The notches in the box plots demonstrate whether medians are different with 95% confidence."

- Different to what? Or do you mean it shows the 95% confidence range of the median?

L486 – "Coefficients of determination (R2) when computing seasonal ACI values"

- What do you mean by this? Is this the correlation coefficient between Nd and the aerosol proxy?

L521 – "Subsequently, one standard deviation from both sides of the seasonal mean defined cut-off points outside of which we assign values as being low and high in each season."

- Could be written more clearly.

L683 – "will struggle for analysing".

L702 – "There were significant changes" – what were the changes?

L705 – "and surface-based aerosol mass concentrations and CCN concentrations (1% supersaturation) are generally highest in JJA and MAM and are at (or near) their lowest values in DJF"

- Surface sulfate aerosol mass concentrations were actually similar in DJF and JJA.

L725 – "by CAO type of conditions" – better as "by conditions associated with CAOs".

---

## Author Response (AR1)

Author Response to Both Referee Comments:

Response: We thank the two reviewers for thoughtful suggestions and constructive criticism that have helped us improve our manuscript. Below we provide responses to reviewer concerns and suggestions in blue font.

**Reviewer 1:**

In the paper "Cloud Drop Number Concentrations over the Western North Atlantic Ocean: Seasonal Cycle, Aerosol Interrelationships, and Other Influential Factors" by Dadashazar et al., the authors investigate cloud droplet number concentrations (Nd) and its influential factors on multiple scales on the basis of diverse observational data sets. The analysis first describes high-resolution aircraft measurements of a flight from the ACTIVATE campaign, and puts these high-resolution measurements into a wider context with the description of the general seasonal cycles of cloud/aerosol/meteorology of the region. The authors go into the details of aerosol size and vertical distributions and calculate aerosol-cloud-interaction (ACI) statistics in different seasons. Potential influential meteorological factors are described in two analyses, using a composite approach by contrasting high and low Nd days, and by the application of a machine learning algorithm. Some of the main findings of the paper are that ACI is generally strongest during DJF, when Nd values tend to be highest, and that high Nd days are shown to feature systematically different meteorology (e.g. stronger continental outflow) when compared to low Nd days.

The topic of the paper is highly relevant to the aerosol/cloud/climate community and of high interest to the readership of ACP. The paper presents a thorough analysis of comprehensive observational data sets that advances the scientific understanding of the observed Nd patterns of the region. The paper is well written and structured and displays high-quality figures. I have only some minor points the authors need to address and some specific remarks that the authors may want to consider. I therefore recommend the manuscript for publication in ACP after minor revisions.

**1 Minor Points**

1.  I think the authors should discuss vertical velocity as one of the main drivers of Nd variability in some more detail. The authors do this to some extent already in the manuscript, but I think it is necessary to point out that this is likely an important factor which is e.g. not provided as an input to the GBRT models. I would suggest that the authors include near-surface wind speed as a proxy for boundary layer turbulence/updrafts in the GBRT, especially since winds seem to be an important factor in the composite analysis.

    Response: We have addressed this comment by including wind speed at 2 m ($Wind_{2m}$) as an input parameter into the GBRT model. We also include near-surface wind direction following the second reviewer's advice.

[revised manuscript text omitted]

Figure S21: Average local accumulated effect (ALE) profiles based on GBRT modeling of the following parameters: (a) relative humidity at 950 hPa ($RH_{950}$), (b) relative humidity at 800 hPa ($RH_{800}$), (c) rain rate, (d) planetary boundary layer height (PBLH), (e) wind speed at 2 m ($Wind_{2m}$), and (f) wind direction at 2 m (wind-dir$_{2m}$). Blue and red profiles represent ALEs of DJF and JJA, respectively. Shaded areas show the ALE ranges stemming from the variability of the obtained models from the cross-validation resampling procedure. Markers on the bottom and top x-axes denote the values of 5[th], 25[th], 50[th], 75[th], and 95[th] percentiles for each input variable; note that the first three markers on the x-axes in panel (c) are very close and thus on top of each other.

2.  In section 2.3 I am missing information on the hyperparameters of the GBRT models and
    how these are tuned during the training/validation phase. This is critical to be able to
    reproduce the results and informative for readers interested in the technical details of the
    model setup. In my opinion, this information could be provided for in a table and may be
    best suited in the supplement, though.

    Response: We addressed this comment by adding a new table in SI file and also adding
    more detailed explanation in the method section (e.g., section 2.3) as follows:
    "Data were split into two sets: training/validation (70%) and testing (30%). Five-fold
    cross-validation was implemented to train the GBRT model using the training/validation
    data. Furthermore, both performance and generalizability of the trained models were
    tested via the aid of the test set, which was not used in the training process.
    Hyperparameters of the GBRT models were optimized through a combination of both
    random and grid search methods. Table S1 shows the list of important hyperparameters
    of the GBRT model and associated ranges tested via random and grid search methods.
    The optimized model hyperparameters can also be found in Table S1. The GBRT models
    were performed using the scikit-learn module designed in Python (Pedregosa et al.,
    2011)."

    **Table S1. Range of model hyperparameters tested during training/validation of the
    GBRT models through a combination of grid and random searches. Final model
    values are also listed in the last column.**

| Model parameter | Range of values tested | Final model values (DJF/JJA) |
| --- | --- | --- |
| Learning rate | 0.001-0.1 | 0.05/0.05 |
| Number of estimators | 100-5000 | 400/400 |
| Maximum depth of a tree | 2-35 | 9/11 |
| Minimum number of smaples to split an internal node | 20-100 | 66/45 |
| Minimum number of smaples at a leaf node | 20-60 | 31/66 |

3.  What is the temporal relationship between the Nd and precipitation data? I believe it
    would be good to a) describe the time of satellite observations in subsection 2.2.1 and the
    precipitation data in 2.2.3 and b) discuss the potential influence of temporal offsets for the
    purpose of analyzing wet scavenging effects. I am also wondering why the authors did
    not chose to use information on precipitation from Cloudsat given they already use A-
    train data.

    Response: We addressed this comment by adding the time of satellite observation in
    section 2.2.1: "Aqua observations used to estimate $N_d$ were from the daytime overpasses
    of the satellite around 13:30 (local time)."

In section 2.2.3 we also described the temporal mismatch between $N_d$ and precipitation and pointed out the potential uncertainties: "It is important to note that we use daily averaged PERSIANN-CDR precipitation and, therefore, there is some temporal mismatch with the daily $N_d$ value from MODIS-Aqua that comes at one time of the day. This can contribute to some level of uncertainty for the discussions based on analyses involving relationships between precipitation and $N_d$."

The lines below in section 4.1 were also updated:

"Furthermore, there was a general reduction in rain on low $N_d$ days for most seasons except SON, with rain enhancement on high $N_d$ days except for DJF (Figure S6); this was unexpected as wet removal was hypothesized to be a reason for reduced $N_d$ for at least the low $N_d$ days. This may be attributed to the rain product being for surface precipitation (and thus not capturing all drizzle) and for all cloud types, including more heavily precipitating clouds deeper and higher than the low-level clouds examined for $N_d$. Another factor potentially contributing to the observed counterintuitive trends is the temporal offset between $N_d$ estimations from MODIS-Aqua and precipitation data from PERSIANN-CDR."

The reviewer raised a good point about why we did not use precipitation data from Cloudsat, which is part of A-train constellation and naturally temporally synchronized with MODIS observations. We had reasons for this choice, with a major one being that Cloudsat does not afford the spatial coverage we desired for our analyses. We had greater flexibility using PERSIANN, however, we note that we are interested in future work to do more detailed types of analyses that could be more catered to the strengths of Cloudsat that are not available from other rain datasets.

4. I think it would beneficial to briefly comment if the vertical distribution of aerosols at nighttime (used here) is excepted to be significantly different from the daytime (rest of the data sets analyzed here).

Response: We added some comments regarding the use of nighttime CALIOP observations in the beginning of section 3.4: "Vertical profiles of aerosol extinction coefficient estimated from CALIOP nighttime observations are shown in Figure 4 for the six sub-domains. Shown also are the seasonally representative planetary boundary layer heights (PBLHs) from MERRA-2, with numerical values of both PBLH and fractional AOD contributions to the PBL and FT in Table 3. Although here we used nighttime observations from CALIOP because of having higher signal to noise ratio than daytime observations, we expect the general seasonal trends discussed here to remain the same regardless of the observation time."

For other work related to another project, we compared extensive CALIOP data for daytime and nighttime conditions for the same region and saw no major qualitative differences.

5. This is just to initiate discussion: In the air-quality community, it is well known that the seasonal cycle of satellite-AOD and near-surface particle concentrations over continental regions frequently show contrasting seasonal cycles (e.g. Koelemeijer et al. 2006, 10.1016/j.atmosenv.2006.04.044). This is caused by effects of PBLH and BL humidity (Stirnberg et al.2018, 10.3390/rs10091353), and can be corrected for to some degree (Arvani et al. 2016, 10.1016/j.atmosenv.2016.06.037). I believe that at least qualitatively there is something to learn from these findings that have implications for the ACI community (and this paper specifically) as well, especially in studies covering continental regions or regions of strong continental outflow. I don't think the authors have to discuss this in their paper, but it may be a useful discussion to have in the ACI community and in my opinion links well to the findings presented here. The authors may chose ignore, comment or discuss this point as they find most useful.

Response: We thank the reviewer for this idea and important point. We aim to consider this line of discussion and thought for potential future lines of work and do not make changes to the current manuscript to address this issue.

**2 Specific Remarks**

l. 58 In my opinion, the word "potentially" does not apply to "enhanced cloud albedo" in the case of increased Nd and constant LWP, but only applies to the latter two points.

Response: Fixed as follows: "It is widely accepted that warm clouds influenced by higher number concentrations of aerosol particles have elevated $N_d$ and smaller drops (all else held fixed), resulting in enhanced cloud albedo at fixed liquid water path (Twomey, 1977), and potentially suppressed precipitation (Albrecht, 1989) and increased vulnerability to overlying air resulting from enhanced cloud top entrainment (Ackerman et al., 2004)."

l. 206 There is a typo: "supermicromemter"

Response: Fixed : "Estimation of supermicrometer particles from FCDP measurements was performed after conducting the following additional screening steps to minimize cloud droplet artifacts:"

l. 209 Please use SI units (760 torrs)

Response: Fixed: "data collected during ACB and BCT legs were excluded. CCN, LAS, CPC, and AMS measurements are reported at standard temperature and pressure (i.e., 273 K and 101.325 kPa) while FCDP and 2DS measurements correspond to ambient conditions."

l. 385 I think Stier (2016, 10.5194/acp-16-6595-2016) is a relevant source that should be cited here.

Response: Thanks for the great suggestions. It is added now: "While previous studies have pointed to the limitations of AOD as an aerosol proxy (e.g. Stier, 2016; Gryspeerdt et al., 2017; Painemal et al., 2020), …"

l. 482 "usually always" - please remove one of them.

Response: Fixed: "Table 4 shows that DJF always exhibits the highest ACI values regardless of the aerosol proxy used, consistent with a stronger aerosol indirect effect in DJF over East Asia."

**Reviewer 2:**
This paper attempts to explain the seasonality of cloud drop number concentrations (Nd) off the coast of the USA and Canada with a focus on explaining why Nd is highest in the DJF season, but the aerosol optical depth (AOD) is lowest in that season. The paper presents some useful analysis and I think it should be published once the concerns have been addressed. However, there are places where the results are not fully explained, some places where there should be a more quantitative analysis and some results that are in the supplementary that should be in the main text. Some of the arguments could also be made more clearly. It seems that aerosol is more efficient at making cloud droplets in DJF, seemingly because of the prevalence of the trade cumulus and stratocumulus conditions (with high low-altitude cloud fractions). However, the paper doesn't quite get the bottom of why this should be – perhaps more discussion on this is warranted although it may be case that we don't quite know yet. Here are some suggestions that might help to get closer to an answer :

- It would be interesting to examine what the important predictors are in the subset of data with high low altitude cloud fractions. E.g., do the aerosol parameters then become equally important between the seasons once we are in the cumulus/stratocumulus regime (particularly CCN and sulphate surface mass)? Is Nd similar in DJF and JJA for the high cloud fraction subset indicating that it is mainly the prevalence of the low cloud conditions in DJF that cause the Nd difference? You could also do a similar analysis for the subset with small low altitude cloud fractions (more typical of JJA perhaps).

    Response: Thank you for the great suggestion. We ran the GBRT models for subsets of data with varying cloud fractions ($CF_{low-liq.}$: 0.2-0.4 and $\geq$ 0.7). The results suggested that in high cloud coverage conditions (i.e., $CF_{low-liq.} \geq 0.7$), sulfate surface mass concentrations were the most important factor regardless of season with $N_d$ showing very similar relationships (and sensitivity) to sulfate surface concentrations in both seasons. In contrast, different aerosol parameters appeared as being more important parameters in DJF and JJA when only data with cloud fractions between 0.2 and 0.4 were included in the GBRT model. In JJA, organic carbon was the most important factor while sulfate was the most important parameter in DJF. It should be noted that for the low cloud fraction model run, $N_d$ exhibited less sensitivity to sulfate in JJA than DJF, which is similar to the results of the original model run including all data points with cloud fraction greater than 0.1.

We added the results of these two sensitivity tests in the SI file and we updated the text in the section 4.2 as follows:

[revised manuscript text omitted]

**Figure S26: Average permutation feature importance of input parameters for (a) DJF and (b) JJA based on GBRT models trained in each season on subsets of data including only samples with low-level liquid cloud fraction between 0.2 and 0.4 (i.e., $0.2 \leq CF_{low\text{-}liq.} \leq 0.4$). Feature importance values were calculated based on using the test set. Error bars exhibit the range of feature importance values stemming from the variability of the obtained models from the cross-validation resampling procedure.**

[Figure]

**Figure S27: Average local accumulated effect (ALE) profiles based on GBRT modeling for**
**surface mass concentrations of the following parameters: (a) dust, (b) organic carbon, (c)**
**sea-salt, and (d) sulfate. ALE profiles were based on GBRT modeling on subsets of data**
**including only samples with low-level liquid cloud fraction between 0.2 and 0.4 (i.e., $0.2 \leq$**
**$CF_{low-liq.} \leq 0.4$). Blue and red profiles represent ALEs of DJF and JJA, respectively. Shaded**
**areas show the ALE ranges stemming from the variability of the obtained models from the**
**cross-validation resampling procedure. Markers on the bottom and top x-axes denote the**
**values of $5^{th}$, $25^{th}$, $50^{th}$, $75^{th}$, and $95^{th}$ percentiles for each input variable.**

[Figure]

**Figure S28: Same as Figure S27 but for the following input parameters: (a) low-level liquid cloud fraction ($CF_{low\text{-}liq.}$), (b) cloud-top effective height of low-level liquid cloud (cloud-top$_{low\text{-}liq.}$), (c) cold-air outbreak (CAO) index, and (d) vertical pressure velocity at 800 hPa ($\omega_{800}$).**

[Figure]

**Figure S29: Same as Figure S27 but for the following input parameters: (a) relative humidity at 950 hPa (RH$_{950}$), (b) relative humidity at 800 hPa (RH$_{800}$), (c) rain rate, (d) planetary boundary layer height (PBLH), (e) wind speed at 2 m (Wind$_{2m}$), and (f) wind direction at 2 m (wind-dir$_{2m}$)."**

- Could it be the case that the Nd retrievals don't work very well and give smaller Nd values when there are no boundary layer clouds (since they are designed to look at PBL liquid clouds)? This could then give higher Nd in the conditions with more PBL clouds (i.e., DJF). There is also likely to be more overlying higher altitude cloud in JJA, which would affect the retrieval of Nd. It would be good to look at the types of clouds and situations in which Nd is being retrieved in JJA – i.e., whether most of the retrievals come from mid-level clouds, or clouds with overlying ice cloud, etc.

Response: We created a new Figure S1 to investigate the potential effects of $N_d$ retrieval errors on the observed seasonal cycle of $N_d$. We also added some text at the end of section 3.2 to describe the results presented in Fig. S1.

"One factor that could drive the seasonal variation in $N_d$ is the unwanted effects of retrieval errors in the estimation of $N_d$ at low cloud coverage conditions. Uncertainty associated with the estimation of $N_d$ from MODIS observation increases as cloud fraction decreases (Grosvenor et al., 2018). This is mainly because of the overestimation of droplet effective radius ($r_e$) in the retrieval algorithm due to the interference of cloud-free pixels and also high spatial inhomogeneity in low cloud coverage conditions that violates horizontal homogeneity assumptions in the retrieval of $r_e$ and $\tau$ from radiative transfer modeling (Zhang et al., 2012; Zhang et al., 2018). To test whether retrieval errors in $N_d$ are the main driver of seasonal trends, Figure S1 shows the seasonal cycle of $N_d$ at various low-level liquid cloud fractions. The results show that as cloud fraction increases the average $N_d$ increases, regardless of season. Perhaps the more important result is that the seasonal trend in spatial maps of $N_d$ remains similar regardless of cloud fraction. This finding is important as confirms that the seasonal cycle in $N_d$ cannot be solely explained by the uncertainties associated with the retrieval of $N_d$ at low cloud fraction."

[Figure]

**Figure S1: Seasonal maps of cloud drop number concentration for different ranges of low-level liquid cloud fraction (CF$_{low\text{-}liq.}$) as follows: (a) $0.1 \leq$ CF$_{low\text{-}liq.}$ $< 0.3$, (b) $0.3 \leq$ CF$_{low\text{-}liq.}$ $< 0.6$, and (c) CF$_{low\text{-}liq.}$ $\geq 0.7$. Gray pixels represent regions without sufficient sample points (less than 10 points) for calculating averages.**

Moreover, the reviewer raised a good point about the potential unwanted effects of high-altitude clouds in estimation of $N_d$. However, this should not be an issue for our analysis as we filter out clouds with cloud top pressure less than 700 hPa; thus, high altitude clouds were automatically removed from our analysis. We added the following information in section 2.1 to clarify this point:

"The CERES-MODIS SSF Level 3 product includes $1° \times 1°$ averaged data according to the cloud top pressure of individual pixels: low (heights below 700 hPa), mid-low (heights within 700–500 hPa), mid-high (heights within 500–300 hPa), and high (heights above 300 hPa) level clouds. For this study, we only use low-cloud averages."

- Can the composite analysis be done with the Cape Cod CCN data? Or can the CCN be used in the ML model? E.g., is it a better predictor than surface sulphate mass?

Response: Unfortunately, it is not feasible to run the models with Cape Cod CCN data for two reasons. First, the data were only available for almost one year, which was not sufficient for our analysis. Second, and maybe even more important, CCN data was a point measurement that provided data at most for only one grid cell over the WNAO region (25° – 50°N and 60° – 85°W), which is not desirable for the analyses presented in this work.

- Can you look at boundary layer decoupling index? Perhaps the high cloud fraction regime in winter is more coupled than the summer regime allowing more efficient transport of the surface aerosol to the clouds?

Response: We agree with the reviewer suggestion that the marine boundary layer in DJF is more likely to be coupled than the conditions in JJA. This is supported by the results presented in Painemal et al. (2021); they showed that surface heat fluxes including both latent and sensible heat fluxes were substantially greater in DJF than JJA. In fact, JJA exhibited the lowest surface heat fluxes among all the seasons, making it more likely to have less turbulence and more prevalence of decoupled marine boundary layer conditions. We addressed this comment by adding the following text at the end of section 3.5:

"The results of this section suggest though that aerosol indirect effects could be strongest in DJF, meaning that $N_d$ values increase more for the same increase in aerosol. Factors that can contribute to higher ACI values in winter than summer include seasonal differences in the following: (i) dynamical processes and turbulent structures of the marine boundary layer; (ii) aerosol size distributions and consequently varying particle number concentrations for a fixed mass concentration; and (iii) hygroscopicity of particles especially as a result of changes in the composition of the carbonacous aerosol fraction. Regarding dynamical processes and the effects of turbulence, Figure 2 in Painemal et al. (2021) shows that heat fluxes (i.e., latent and sensible fluxes) are strongest (lowest) in the winter (summer) over the WNAO. The greater heat fluxes in DJF can contribute to more turbulent and coupled marine boundary layer conditions in winter than summer, presumably resulting in more efficient transport and activation of aerosol in the marine boundary layer leading to higher ACI values. Forthcoming work will probe this issue in greater detail."

- It seems that the offshore flow prevalent in DJF (cf. southeasterlies from the open ocean in JJA) might also play some role in determining Nd since it may transport more CCN from the continent? Are there more measurements that can help to elucidate whether this may be the case (e.g., aerosol size distribution data or additional CCN data from the Cape Cod data)? It might be good to include the wind direction as a predictor in the ML?

Response: We already tried to answer this question by looking at Cape Cod data and CALIPSO results presented in section 3.4. Another place where we looked at the effect of size was the section 3.3 where we incorporated AI values rather than AOD in ACI estimations.

Based on this comment we also included wind direction at 2 m (Wind-dir$_{2m}$) as an input parameter of the GBRT models. Interestingly, we could not find a clear relationship between $N_d$ and wind direction. We refer the reviewer to the previous comments to see the results of adding wind direction as an input parameter. This idea though is important and something that warrants additional investigation with more detailed data.

• Other possible reasons for the DJF aerosols (or aerosols when there is lots of PBL cloud) being more efficient at making droplets could be discussed. E.g., could the aerosol be more hygroscopic in these conditions, could there be higher number concentrations (since here you mainly consider mass concentrations). These difference may be related to the different aerosol sources due to the different wind direction (see previous point).

Response: We addressed this comment by adding the following text at the end of section 3.5: "The results of this section suggest though that aerosol indirect effects could be strongest in DJF, meaning that $N_d$ values increase more for the same increase in aerosol. Factors that can contribute to higher ACI values in winter than summer include seasonal differences in the following: (i) dynamical processes and turbulent structures of the marine boundary layer; (ii) aerosol size distributions and consequently varying particle number concentrations for a fixed mass concentration; and (iii) hygroscopicity of particles especially as a result of changes in the composition of the carbonacous aerosol fraction. Regarding dynamical processes and the effects of turbulence, Figure 2 in Painemal et al. (2021) showed that heat fluxes (i.e., latent and sensible fluxes) are strongest (lowest) in the winter (summer) over the WNAO. The greater heat fluxes in DJF can contribute to more turbulent and coupled marine boundary layer conditions in winter than summer, presumably resulting in more efficient transport and activation of aerosol in the marine boundary layer leading to higher ACI values. Forthcoming work will probe this issue in greater detail."

• Why did you not include AOD, speciated AOD, speciated boundary layer AOD, etc., in the ML model so that their impact relative to sulphate surface mass, etc. can be quantified?

Response: We originally included these parameters in our early analyses but for simplicity (i.e., to cut down on having too many figures) we decided to only present the results for surface mass concentration as we thought the latter parameters were more relevant because they should be more closely linked to the aerosol level near the cloud base. We ran the model again though using speciated AOD values rather than surface mass concentrations. The general results are quite similar to the versions currently in the draft with some changes in the importance of parameters. Moreover, ALE plots are very similar to the versions in the main draft. As such, we decided to not include these results in the main draft:

[Figure]

**Same as Figure 13 of the main draft but instead of surface mass concentrations,**
**speciated AODs were included as input parameters to GBRT models.**

[Figure]

**Same as Figure 14 of the main draft but instead of surface mass concentrations,**
**speciated AODs were included as input parameters to GBRT models.**

[Figure]

**Same as Figure 15 of the main draft but instead of surface mass concentrations,**
**speciated AODs were included as input parameters to GBRT models.**

[Figure]

Same as Figure S21 of the main draft but instead of surface mass concentrations,
speciated AODs were included as input parameters to GBRT models.

Also some more line specific comments :-

Section 3.1 – I'm not sure how well this section works where it is, or how useful the aircraft analysis is for the main conclusions of the paper. Maybe it would be better placed at the end of all the other results in order to help highlight some of the issues raised in the rest of the paper?

> Response: We thought about this at great length when designing the very first draft we originally submitted. We felt it was an exciting and compelling opener into the wide gradient in $N_d$ over the study region using a high quality (and "hard to get") airborne field dataset. We wanted to raise points from the airborne case study to motivate the general topic we investigate throughout the rest of the draft. We still feel the section is suitable where it currently is placed.

> L318 – Is it possible to calculate an approximate activation diameter for 0.43% supersaturation given the other aircraft measurements? This would make it easier to compare to the Dp>10 and Dp>3um data.

Response: This is a good idea but there are some challenges and overall we do not feel it is needed for the overall story of the paper. More specifically, we would need to stitch together size distribution data from various instrument like SMPS and LAS that measure different types of diameters (e.g., aerodynamic diameter and electrical mobility diameter). The time resolution of the SMPS is much longer than that of the LAS and, importantly, SMPS data were not available in such a way to represent the whole flight segment under consideration.

L381 – "Consequently, humidity effects on remotely sensed aerosol parameters cannot alone explain the dissimilar seasonal cycle of Nd and AOD, but can plausibly contribute to some extent."
- You haven't proved this quantitatively. Can you do a calculation of how much impact the RH difference would have?

Response: Without knowing the exact composition of aerosol it is difficult to quantify the effects of RH on extinction profiles. Therefore, we change the words to address the point reviewer raised here:

"Consequently, humidity effects on remotely sensed aerosol parameters are less likely to be sole explanation of the dissimilar seasonal cycle of $N_d$ and AOD, but can plausibly contribute to some extent."

L411 – "but do not contribute significantly to number concentration as demonstrated clearly by airborne observations (Figure 1)."
- It's not clear which part of Fig. 1 demonstrates this? Can you explicitly point this out?

> Response: We addressed this comment by adding clarifying words:

 "but do not contribute significantly to number concentration as demonstrated clearly by
 airborne observations (i.e., $FCDP_{>3\mu m}$ timeseries shown in Figure 1d)."

L413 – "This is supported in part by how DJF is marked by the highest fractional AOD
contribution from the PBL (59 – 72%) where sea salt is concentrated. In contrast, JJA has the
lowest fractional AOD contribution from the PBL (11.3 – 52.6%)."
- But this could also indicate higher CCN concentrations in the PBL in DJF perhaps due to the
aerosol being more trapped there than in summer?

 Response: Here we only made a speculation, which we think is more likely given other
 observations. To address the comment, we mention the other possibility based on the
 reviewer suggestion:
 "It is also possible that the higher fractional AOD contribution from the PBL in winter
 partly owes to aerosol particles being more strongly confined to the PBL as compared to
 the summer."

Figure 3 – It looks like here the PBL AOD would be higher in DJF than for JJA for many of
the regions. It would be good to quote the PBL AOD values in Table 3. You need to describe
these results in more detail - the enhanced PBL values in DJF are an important point to
describe even though it seems that it doesn't explain the Nd seasonality.

 Response: We already mentioned the percentage of AOD that comes from PBL and FT in
 Table 3. We do not believe it is necessary to report these additional numbers.

L447 – "We next compare MERRA-2 speciated aerosol concentrations at the surface (Figure S2)
to those of speciated AOD (Figure S1)."
- I think that this information is very important and should not be in the supplementary
since whether the PBL aerosol mass concentrations (or ideally CCN number
concentrations, but I think they are not available from MERRA?) are lower in DJF
compared to JJA is a key part of the analysis regarding why the DJF AOD is lower and
yet the Nd higher compared to JJA. Indeed, Fig. S2 suggests that surface sulfate mass
concentration is about the same in DJF and JJA despite the AOD difference (and sulfate
AOD is also higher in JJA than DJF according to MERRA). Of course we might expect
sulfate mass concentrations to be even higher in DJF than in JJA if it was to explain the
Nd difference, but it does suggest that part of the issue is that AOD is vertically
integrated and is not just for the PBL.
Response: We made the requested change.
- Figs. S1 and S2 also suggest that near the coast sulfate AOD dominates over sea-salt,
which argues against the higher observed (Fig. 3) PBL extinction values in DJF being
due to sea-salt (as argued e.g., L413 and L707).

 Response: A closer inspection of the former Figure S2 though shows very different
 number scales for sea salt versus sulfate that should not be ignored. As that figure quantifies surface mass concentrations (relevant to PBL), we feel it still supports sea being a dominant contributor to PBL aerosol mass and optical depth.

- Although sea-salt surface mass concentrations are higher than sulfate. Do you have speciated profiles of extinction from MERRA? These could be used to quantify the effect of sea-salt on the PBL AOD.

Response: We looked for such data but could not locate it as part of the standard MERRA-2 data products.

- What seems a bit strange given that DJF and JJA have similar surface SO4 in MERRA is that the Cape Cod observations show lower CCN concentrations in DJF. It would also be good to discuss this a bit more along with potential caveats. You mention that the 1% supersaturation at which the CCN are measured is quite high and would be counting fairly small aerosol particles – I think it would be good to show the lower supersaturations that you say are available. Or at least check whether the DJF values of these are also lower than in JJA (data permitting). Is there any other CCN data down the east coast of US since it would be very useful here. Or do you have observed aerosol size and composition measurements that might help determine whether the supersaturation has a big effect and whether there really are fewer CCN in DJF?

Response: Regarding CCN data at lower supersaturation, unfortunately the temporal coverage of CCN data at Cape Cod was not good enough to give a full seasonal profile at lower supersaturation. About other datasets along the East Coast we do not have those at our disposal to use. These are all good avenues of future research.

      o Also, how representative is the 1-year of data likely to be? Could the interannual variability be large enough to make that result uncertain?

Response: This is a hard question to robustly answer without any uncertainty without having the data. It is our anticipation that there would not be significantly high year-to-year variability to change the story. We did not feel this comment needed additional revision.

      o It could also be that MERRA is doing a poor job of representing the sulphate mass concentration.

Response: Answering such a question is outside to scope of our study.

      o Finally, it would be good to quantify how likely it is that we can have a similar surface sulphate mass concentration, but different CCN (using observations or a more sophisticated model perhaps).

Response: It is our opinion that this type of question is more geared towards future work and outside the scope of this current paper's objectives.

Typos / grammar etc.

L252 – "The ALE value of feature S" – it's not clear here what you mean by "feature". The symbols for this equation have also not all been explained. What does subscript c refer to? What are xs, xc, zs? "The value of $f_{s,ALE}(x_s)$ can be viewed as the difference between the model's response at xs and the average prediction." – this is not very clear – I think this explanation needs to be clearer for interpreting the later figures.

Response: We clarified these variables as follows:

"ALE plots illustrate the influence of input variables on the response parameter in ML models. The ALE value for a particular variable S at a specific value of $x_s$ (i.e., $f_{s,ALE}(x_s)$) can be calculated as follows:

$$f_{s,ALE}(x_s) = \int_{z_{0,1}}^{x_s} \int_{x_c} f^s(z_s, x_c) \, P(x_c|z_s) dx_c dz_s - constant \qquad (2)$$

where $f^s(z_s, x_c)$ is the gradient of model's response with respect to variable S (i.e., local effect) and $P(x_c|z_s)$ is the conditional distribution of $x_c$ where C denotes the other input variables rather than S and $x_c$ is the associated point in the variable space of C. $z_{0,1}$ is chosen arbitrarily below the smallest observation of feature S (Apley and Zhu, 2020)."

L288 – the flight leg abbreviation "BCT1" is stated here without definition. It would be good to have introduced the different flight legs being described before this with reference to Fig. 1.

Response: We added a line referring readers to the caption of Fig.1 for the definition of different legs: " Sea surface temperatures were $6 - 9°C$ near the coast during the descent and Min. Alt. 1 leg (readers are referred to Fig. 1's caption for the definition of different legs)…"

L305 – "ranged" -> "ranged from".

Response: Fixed as follows: "$N_d$ values from the FCDP ranged from a maximum value of 1298 $cm^{-3}$ …"

L307 – "which is a fairer comparison with the ACB1 leg" – fairer than what?

Response: Fixed as follows: "The minimum $N_d$ value in the ACB3 leg was 85 $cm^{-3}$ (34.11° N, 72.80° W), which is a fairer comparison to the ACB1 leg as compared to the BCT1 leg in terms of being closer to cloud base."

L311 – "there was a significant offshore gradient in LAS submicrometer particle number concentration and AMS non-refractory aerosol mass, ranging from 424 cm-3 and 5.60 µg m-3 (from BCB1) to 21 cm-3 and 0.32 µg m-3 (from BCB3), respectively; these values are based on times of the maximum and minimum LAS concentrations during the BCB1 and BCB3 legs,
respectively."
- This sentence was a little confusing and could be made clearer I think.

Response: We revised the text: "there was a significant offshore gradient in LAS submicrometer
particle number concentration and AMS non-refractory aerosol mass, ranging from as high as
424 cm$^{-3}$ and 5.60 µg m$^{-3}$ (during BCB1) to as low as 21 cm$^{-3}$ and 0.27 µg m$^{-3}$ (during BCB3)."

L320 – "There was a slighter gradient in particle concentrations with Dp > 3 µm (most likely sea
salt) between the same two points of maximum and minimum LAS concentration in BCB1 and
BCB3 legs, respectively: 0.26 cm-3 to 0.11 cm-3."
- This could be written more clearly.

Response: Revised this general section for clarity: "For the duration of the flight portion shown
in Figure 1, supermicrometer concentrations varied over two orders of magnitude (0.002 – 0.51
cm$^{-3}$) and expectedly did not exhibit a pronounced offshore gradient as it is naturally emitted
from the ocean."

Fig. 4 – "The notches in the box plots demonstrate whether medians are different with 95%
confidence."
- Different to what? Or do you mean it shows the 95% confidence range of the median?

Response: They show whether medians are different from each other with 95% confidence. We
updated the caption as follows: **"The notches in the box plots demonstrate whether medians
are different from each other with 95% confidence. Boxes with notches that do not overlap
with each other have different medians with 95% confidence."**

L486 – "Coefficients of determination (R2) when computing seasonal ACI values"
- What do you mean by this? Is this the correlation coefficient between Nd and the
aerosol proxy?

Response: We clarified this as follows: "Coefficients of determination ($R^2$) for the linear
regression between ln($N_d$) and ln($\alpha$) when computing seasonal ACI values were generally low ($\leq$
0.30), with spatial maps of $R^2$ and data point numbers in Figure S2."

L521 – "Subsequently, one standard deviation from both sides of the seasonal mean defined cut-
off points outside of which we assign values as being low and high in each season."
- Could be written more clearly.

Response: Revised text: "We assign values as being low in each season if they are less than one standard deviation below the seasonal value; conversely, high values are those exceeding one standard deviation above the seasonal mean."

L683 – "will struggle for analysing".

Response: Revised text: "Satellite remote sensing studies of aerosol-cloud interactions presumably will be more challenging in winter periods versus the summer with regard to the spatial and temporal mismatch between cloud and aerosol retrievals."

L702 – "There were significant changes" – what were the changes?

Response: We added the following lines in response to this comment: "These changes included a sharp decrease in aerosol number concentration, a decrease in mass fraction of sulfate in droplet residual particles, and an increase in mass fraction of organic and chloride of droplet residual particles moving offshore."

L705 – "and surface-based aerosol mass concentrations and CCN concentrations (1% supersaturation) are generally highest in JJA and MAM and are at (or near) their lowest values in DJF"

- Surface sulfate aerosol mass concentrations were actually similar in DJF and JJA.

Response: Revised text: "$N_d$ is generally highest (lowest) in DJF (JJA) over the WNAO but aerosol parameters such as AOD, AI, surface-based aerosol mass concentrations for most species, and CCN concentrations (1% supersaturation) are generally highest in JJA and MAM and are at (or near) their lowest values in DJF."

L725 – "by CAO type of conditions" – better as "by conditions associated with CAOs".

Response: Fixed: "…which is assisted in large part by conditions associated with CAOs such as high cloud fraction and high CAO index."

References:

Painemal, D., Corral, A. F., Sorooshian, A., Brunke, M. A., Chellappan, S., Gorooh, V. A., Ham, S., O'Neill, L., Smith Jr., W. L., Tselioudis, G., Wang, H., Zeng, X., and Zuidema, P.: An Overview of Atmospheric Features Over the Western North Atlantic Ocean and North American East Coast – Part 2: Circulation, Boundary Layer, and Clouds, J Geophys Res-Atmos, 10.1029/2020JD033423, 2021.